# Internal water channel formation in CXCR4 is crucial for $G_i$-protein coupling upon activation by CXCL12

Chun-Chun Chang[1,2,5], Je-Wen Liou[3,5], Kingsley Theras Primus Dass[4], Ya-Tzu Li[3], Shinn-Jong Jiang[3], Sheng-Feng Pan[3], Yu-Chen Yeh[3,4] & Hao-Jen Hsu [3,4✉]

Chemokine receptor CXCR4 is a major drug target for numerous diseases because of its involvement in the regulation of cell migration and the developmental process. In this study, atomic-level molecular dynamics simulations were used to determine the activation mechanism and internal water formation of CXCR4 in complex with chemokine CXCL12 and $G_i$-protein. The results indicated that CXCL12-bound CXCR4 underwent transmembrane 6 (TM6) outward movement and a decrease in tyrosine toggle switch by eliciting the breakage of hydrophobic layer to form a continuous internal water channel. In the GDP-bound $G_{\alpha i}$-protein state, the rotation and translation of the α5-helix of $G_{\alpha i}$-protein toward the cytoplasmic pocket of CXCR4 induced an increase in interdomain distance for GDP leaving. Finally, an internal water channel formation model was proposed based on our simulations for CXCL12-bound CXCR4 in complex with $G_{\alpha i}$-protein upon activation for downstream signaling. This model could be useful in anticancer drug development.

[1] Department of Laboratory Medicine, Hualien Tzu Chi Hospital, Buddhist Tzu Chi Medical Foundation, Hualien 97004, Taiwan. [2] Department of Laboratory Medicine and Biotechnology, College of Medicine, Tzu Chi University, Hualien 97004, Taiwan. [3] Department of Biochemistry, School of Medicine, Tzu Chi University, Hualien 97004, Taiwan. [4] Department of Life Sciences, Tzu Chi University, Hualien 97004, Taiwan. [5]These authors contributed equally: Chun-Chun Chang, Je-Wen Liou. ✉email: hjhsu32@mail.tcu.edu.tw

XCL12 is a pleiotropic chemokine commonly present in numerous tissues and acts as a chemoattractant, playing a crucial role in inflammation and immune surveillance of tissues[1]. CXCL12 is the only known endogenous ligand for CXC chemokine receptor 4 (CXCR4), also known as fusin or cluster of differentiation 184 (CD184). CXCR4 is composed of 352 amino acids and has a molecular weight of 48 kDa. Cells expressing CXCR4 migrate along the CXCL12 concentration gradient and are involved in diverse physiological functions and organ development[2]. The CXCL12/CXCR4 axis regulates various cellular behaviors, such as cell migration, adhesion, and invasion[3].

The crystal structure of CXCR4 lacking the N-terminus was solved in 2010 with antagonists bound[4]. Furthermore, the NMR structure of CXCL12 bound to the N-terminus of CXCR4 was released to assess their critical interactions[5]. Studies have also reported that CXCL12 binding to CXCR4 follows a two-site binding mechanism, which suggests that binding at Site 1 occurs when the N-loop (RFFESH) of CXCL12 interacts with the N-terminus of CXCR4 for the initial binding, whereas binding at Site 2 occurs when the N-terminus of CXCL12 interacts with the top groove of the transmembrane (TM) helices of CXCR4[5–7]. Moreover, CXCR4 binding to its chemokine induces the activation of various intracellular (IC) signaling transduction pathways and downstream effectors that mediate cell survival, proliferation, chemotaxis, migration, and adhesion through the transmembrane helices[8]. Substantial conformational changes in G-protein-coupled receptor (GPCR) TM helices have been demonstrated to mediate signaling upon activation[9,10]. Conformational changes occur in the transmembrane and IC regions of CXCR4 following binding of chemokine CXCL12 to CXCR4, which act as signals for the heterotrimeric inhibitory G-protein ($G_i$) binding. Trimeric $G_i$-protein couples with the activated receptor, which causes the $G_{\alpha i}$ subunit to undergo a conformational change that promotes the domain separation in $G_{\alpha i}$-protein and the exchange of GDP to GTP for downstream signaling[11].

The structures reported[4,12] to date have been in an inactive state, and thus information regarding the complete conformational change process of CXCR4 from the inactive to active state is still lacking. Experimental data provided extensive information regarding the interactions between CXCL12 and CXCR4 but did not address the interactions between the activated CXCR4 and G-protein. Although studies have displayed the complex structures of nucleotide-free G-protein bound to GPCR[13–15], the detailed dynamic process of how activated GPCR induces GDP/GTP exchange inside the G-protein remains unclear. The complete atomic-level signaling network from agonist CXCL12 binding to G-protein through CXCR4 is still not available. To address these questions, we used atomic-level molecular dynamics (MD) simulations to clarify the activation mechanism and internal water channel formation of receptor CXCR4 in complex with chemokine CXCL12 and G-protein. Other simulation systems, such as a small-molecule antagonist isothiourea derivative (IT1t) bound to CXCR4 and CXCL12 bound to mutant CXCR4 (mCXCR4), were also used to perform microsecond-scale MD simulations for comparison. The $G_i$-protein was then coupled to the cytoplasmic region of CXCL12-bound CXCR4 to investigate the dynamics of conformational changes of the G-protein. Our simulations suggested that electrostatic interactions may dominate the binding of CXCL12 to receptor CXCR4. The CXCL12-bound CXCR4 undergoes conformational changes in the TM region, which in turn enables the internal waters to flow through the TM region by eliciting the breakage of hydrophobic layer (HL). When GDP-bound $G_{\alpha i}$-protein couples with the CXCL12-bound CXCR4, the increase in interdomain distance and the rotation and translation of the α5-helix of $G_{\alpha i}$-protein toward the cytoplasmic pocket of CXCR4 may cause the unbinding of GDP

from the nucleotide-binding site of $G_{\alpha i}$-protein. Therefore, on the basis of our simulations, we proposed an internal water channel formation model for CXCL12-bound CXCR4 in complex with $G_{\alpha i}$-protein upon activation for downstream signaling. CXCR4 has been demonstrated to be a prognostic marker associated with numerous cancers, such as breast, prostate, lung, and colon cancers, where it promotes metastasis, angiogenesis, and tumor growth or survival; therefore, our results provide valuable information for the understanding the downstream signal transduction process of the CXCL12–CXCR4–$G_{\alpha i}$ tricomplex, which could prove useful in the development of anticancer and antimetastatic drugs.

## Results and discussion

**Electrostatic interactions dominate the binding of chemokine CXCL12 to receptor CXCR4.** To explore the ligand binding to receptor CXCR4, various ligands docked to receptor CXCR4 were studied under three circumstances, namely CXCL12 docked into the TM region of CXCR4, small-molecule antagonist IT1t redocked to receptor CXCR4, and CXCL12 docked to mCXCR4 (L244$^{6.40}$P and L246$^{6.42}$P). It was found in previous mutagenesis experiments that the mutations at these two positions did not affect the CXCL12 binding, but noticeably reduced calcium flux[9,16], causing the inactivation of CXCR4. Thus, the proline mutation in this region can eliminate the downstream signaling. The docking results are presented in Supplementary Tables S1 and S2 and Fig. 1, Supplementary Figs. S1a and S2b. Although the difference of RDOCK energy between different docking poses was not particularly large, the binding conformations of CXCL12 docked to CXCR4 system were noticeably different, and only the docking pose1 displayed the conformation of CXCL12 embedded into CXCR4 (Supplementary Fig. S1b). We selected the pose1 as the preferable pose for CXCL12 binding to CXCR4, on the basis of findings from experimental[6,9,17] and simulation[7,18] studies. The preferred pose determined was used for further MD simulations. The predicted binding interface for CXCL12–CXCR4 revealed substantial overlap with the binding of the N-terminus of CXCR4 to the CXCL12 structure (PDB: 2N55; Supplementary Fig. S1c)[17]. The superposition RMSD of the two structures was 3.2 Å. The docked CXCL12−CXCR4 complex structure was superposed to the model proposed by Ziarek et al.[17], which had an RMSD of 4.5 Å, indicating that these two models were comparable (Supplementary Fig. S1d). Surface charge distribution maps demonstrated that the N-terminus and N-loop of CXCL12 with more positively charged residues (K1, R8, and R12) were embedded into the extracellular (EC) region of CXCR4 with stronger negative electric fields (D10$^{N-term}$, E14$^{N-term}$, E15$^{N-term}$, E31$^{1.25}$, E32$^{1.26}$, E179$^{ECL2}$, D181$^{ECL2}$, D182$^{ECL2}$, D187$^{ECL2}$, D193$^{5.32}$, D262$^{6.58}$, and E288$^{7.39}$), which suggested that electrostatic interactions may play a major role in the binding of CXCL12 to CXCR4 (Fig. 1). This result was consistent with previous findings[7,9] and also resembled other chemokine-receptor bindings (CXCL8–CXCR1)[19,20]. CXCL12 bound to mCXCR4 also exhibited similar results of electrostatic interactions dominating the binding (Supplementary Fig. S1a).

Previous studies have suggested a two-site binding model for CXCL12 binding to CXCR4[6,7,9,18,21] in which the N-loop, β-sheet, and 40s-loop of CXCL12 interact with the extracellular region of CXCR4 (Site 1) and the N-terminus of CXCL12 buries into the TM region of CXCR4 with electrostatic interactions to trigger the activation and signaling of CXCR4 (Site 2). For the CXCL12-bound CXCR4 system, after 1.8-μs MD simulations, the N-terminus of CXCL12 was still buried deep in CXCR4 and occupied the entire EC region and most of the TM region. The interactions of the RFFESH loop of CXCL12 with the N-terminal domain of

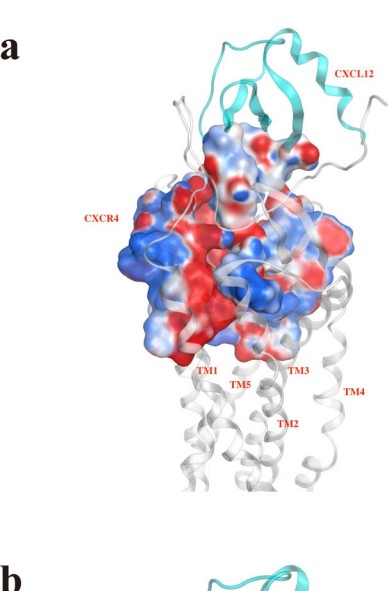

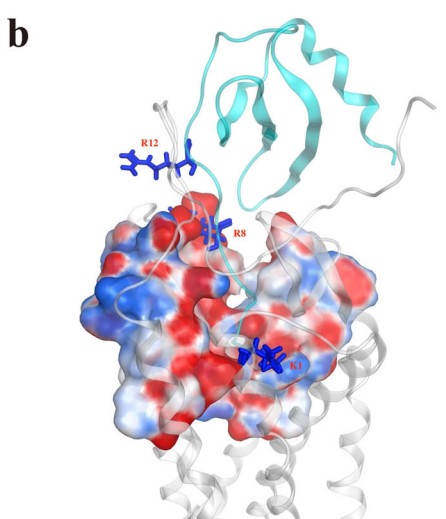

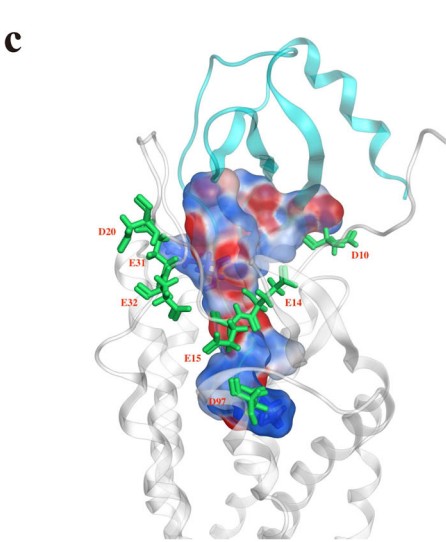

**Fig. 1 CXCL12 binding to CXCR4 and surface charge distribution.** The most preferable docking pose was determined on the basis of molecular docking results and previous experiments. The surface charge distribution was calculated using the Poisson–Boltzmann equation. The complex is represented as a ribbon with agonist CXCL12 colored cyan and receptor CXCR4 colored gray. Blue color corresponds to positive and red color to negative electrostatic potential. Residues around the binding interface are labeled and depicted as sticks; blue is for CXCL12, and green is for receptor. **a** Surface charge distribution around the binding interface for CXCL12 binding to CXCR4. **b** Surface charge distribution of CXCR4 inside the receptor region presented for clarity are more negative charge potential. Positively charged residues of the N-terminus of CXCL12 are labeled to show interactions with CXCR4. **c** Surface charge distribution of the N-terminus of CXCL12 showing more positive charge potential. Negative charged residues of the extracellular region of CXCR4 are labeled to show interactions with CXCL12.

**Site 2 binding triggers G-protein signaling.** The heatmap of these interactions over time also indicated that most of these interactions were maintained under 4.0 Å (probability >50%) during the simulation time (Supplementary Fig. S2a). Simulations were also performed for the replicate CXCL12−CXCR4 system and another CXCL12−CXCR4 model proposed by Floudas et al.[18]. The results showed that tyrosine toggle switch and RMSD profiles over time were comparable to our model (Supplementary Fig. S1e, f). The tyrosine toggle switch was determined by measuring the distance between Y219[5.58] and Y302[7.53]. Previous studies have indicated that the two residues are closer during GPCR activation[9,22]. The preferable docking pose of antagonist IT1t to CXCR4 was similar to the solved crystal structure of CXCR4 with IT1t bound and the RMSD of superposition of the two structures was 0.59 Å (Supplementary Fig. S2b)[4]. The IT1t-bound CXCR4 structure embedded into a hydrated POPC lipid bilayer displayed stable RMSD values in fluctuations around 0.45 nm after 1.5-μs MD simulations. The IT1t−CXCR4 system may maintain the inactive state during the simulation time compared with the initial crystal structure system (Supplementary Figs. S1e and S2c). Mutagenesis experiments have suggested that mutant CXCR4 (L244[6.40]P and L246[6.42]P) does not affect the CXCL12 binding, but does eliminate the downstream signaling[4,9,21]. The simulations indicated that CXCL12-bound mCXCR4 exhibited similar binding interactions to the CXCL12-bound CXCR4 system during the 1.5-μs MD simulations. However, the down half-helix of TM6 with two mutated residues moved dynamically, and the superposition of the structures at different time frames (0, 500, 1000, and 1500 ns) showed the inward tilt of 5.1 Å without affecting the ligand binding (Supplementary Fig. S2d). The inward movement of the lower half of TM6 may diminish the cytoplasmic region of CXCR4. This movement is suboptimal for G$_i$-protein binding and inhibits the downstream signaling, which is consistent with the previous experiments[4,16]. The CXCL12−mCXCR4 complex may also be in an inactive state.

**Conformational changes of receptor CXCR4 as agonist CXCL12 binding.** Studies have reported that GPCRs undergo substantial structural changes upon activation, to associate with G-protein for downstream signaling, especially at TM5 and TM6, but also at TM3 and TM7 in certain cases[9,23,24]. The conformational change of CXCR4 induced by CXCL12 binding during long-term MD simulations was illustrated by B-factor (Fig. 3a). The EC region of CXCR4 was mobile because of its interactions with CXCL12, and the lower halves of TM5, 6, and 7 exhibited more fluctuations when coupled with G-protein. However, the B-factor

CXCR4 and C9 of CXCL12 with E277[7.28] and H281[7.32] of CXCR4 were stable during the MD simulations, which was consistent with studies that indicated that Site 1 is the recognition site for CXCL12 binding to CXCR4 (Fig. 2). Moreover, the interactions between the K1 of CXCL12 and the E288[7.39] of CXCR4 and between the S4, S6, and R8 of CXCL12 and the D187[ECL2], D262[6.58], and E277[7.28] of CXCR4 were predominant, which supports the hypothesis that

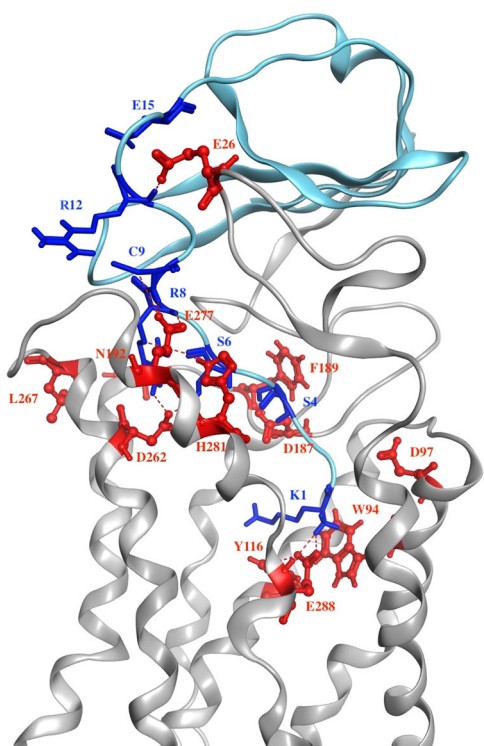

**Fig. 2 Chemokine CXCL12 interaction with receptor CXCR4 at the end of simulation time.** The complex is represented as a ribbon with chemokine CXCL12 colored in light blue and receptor CXCR4 colored gray. Residues around the binding interface are labeled and depicted as sticks; blue is for CXCL12, and red is for receptor. The interactions presented are consistent with previous experiments in which Site 1 is the recognition for CXCL12 binding to CXCR4, and Site 2 binding triggers G-protein signaling.

of other systems exhibited less mobility in the TM helices compared with the CXCL12–CXCR4 system and other regions might have more mobility than TM regions, which indicated that these systems might have less conformational changes in TM helices (Supplementary Fig. S3a–c). For the CXCL12-bound CXCR4 system, the superposition of CXCR4 frames at various simulation time points demonstrated that the IC part of TM5 moved to TM6, whereas the IC region of TM6 vibrated and moved outward. To clearly represent the outward movement of TM6, we superposed the CXCR4 structure frames at different simulation time (0, 500, 1000, 1500, and 1800 ns) and measured the $C_\alpha$ atom movement of the intracellular end residue of TM6 (K234[6.30]). The outward movement of the intracellular end residue of TM6 between the initial and final frames was measured as 5.5 Å (Fig. 3b). This system may report slightly less outward movement compared with other GPCR systems (motion ranges from 6 to 14 Å)[10]. However, our simulation of the CXCL12−CXCR4 system indicated that the outward movement increased activation, which contrasted with other GPCR−$G_{\alpha i}$ systems (motion ranges about 6 to 8 Å)[14,15]. The movement and tilt of TM6 were relatively similar in the apo CXCR4 system; however, interestingly, TM6 moved inward in the mutant CXCR4 system, indicating that the inward movement of TM6 may reduce the cytoplasmic region and inhibit $G_i$-protein binding (Supplementary Fig. S3d, e). Three $C_\alpha$ atoms of residues (I245[6.41], P254[6.50], and G258[6.54]) were selected to measure the kink angle of TM6 (the angle between up half and down half of TM6). The kink angle of TM6 with time for various simulation systems also indicated that larger kink angles (150°~160°) were observed in apo and IT1t-bound CXCR4 systems similar to the inactive CXCR4 crystal structure (150°, PDB: 3ODU), while

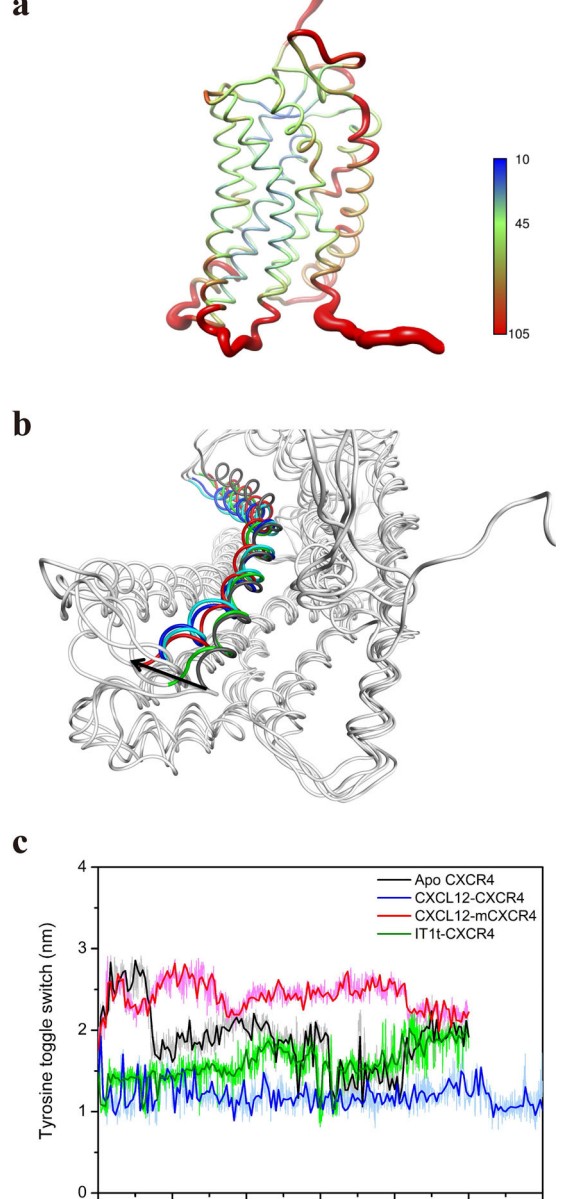

**Fig. 3 Conformational changes and molecular switches of receptor CXCR4 during MD simulations. a** The conformational change of CXCR4 induced by CXCL12 binding during MD simulation is indicated by B-factor of $C_\alpha$ atoms. In the color spectrum, red indicates more flexibility, and blue represents more rigidity. **b** Superposition of CXCR4 frames at various simulation time points shown as intracellular perspective view. CXCR4 frames are represented as gray ribbons; gray is for TM6 at the initial time, blue is for TM6 at 500 ns, cyan is for TM6 at 1000 ns, green is for TM6 at 1500 ns, and red is for TM6 at 1800 ns. **c** Tyrosine toggle switch. The distance between the oxygen atoms of the side chains of Y219[5.58] and Y302[7.53] was measured throughout MD simulation. Apo CXCR4, CXCL12-bound mCXCR4, CXCL12, and antagonist IT1t-bound CXCR4 are represented using black, red, blue, and green lines, respectively.

smaller kink angles (135°~145°) were found in CXCL12-bound CXCR4 and CXCL12−CXCR4−empty $G_{\alpha i}$ systems close to the active $\beta_2AR$−$G_s$ complex structure (130°, PDB: 3SN6). The negative kink angle for CXCL12-bound mCXCR4 system meant inward tilt of down half of TM6. Larger down half tilt movement

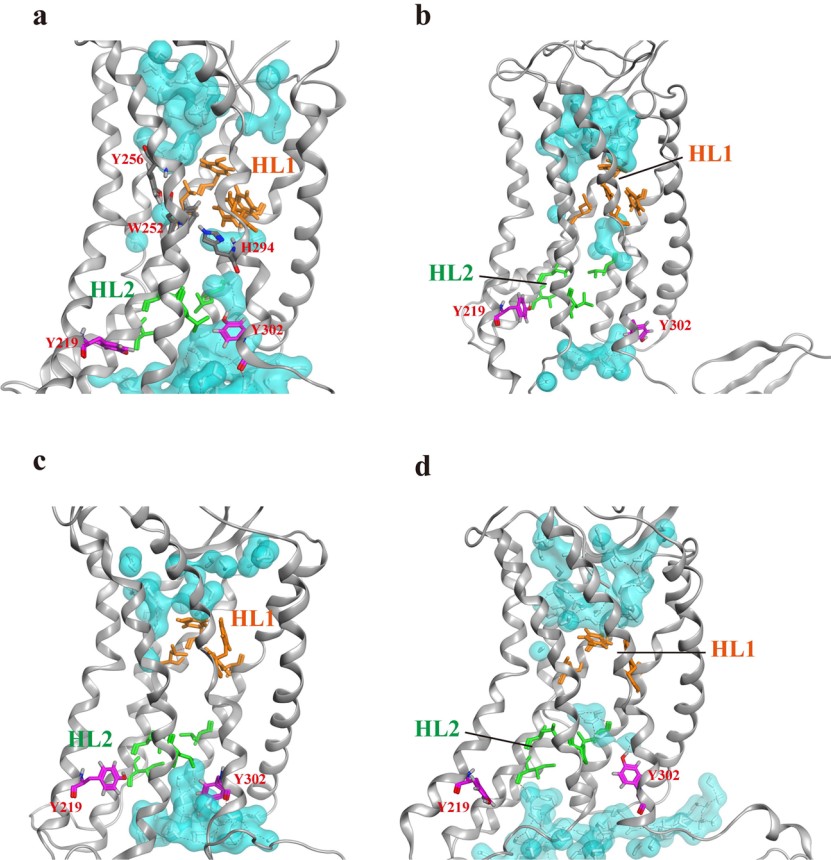

**Fig. 4 Internal water distribution within the transmembrane region of various CXCR4 systems. a** CXCL12-bound CXCR4; **b** apo CXCR4; **c** IT1t-bound CXCR4; **d** CXCL12-bound mutant CXCR4. Water molecules are drawn in both white sticks and cyan for the molecular surface. The side chains of HL1 are illustrated by orange sticks, and the side chains of HL2 are indicated by green sticks. The side chain of residue Y302[7.53] and Y219[5.58] are represented as purple sticks, and Y256[6.52], W252[6.48], and H294[7.45] are labeled and depicted as gray sticks. Hydrogen bonding networks are also shown as red dotted line.

of TM6 means the smaller kink angle of TM6 (Supplementary Fig. S3f).

Although GPCR activation is based on the action of several molecular switches[25,26], only the tyrosine toggle switch was observed in our long-term MD simulations for CXCR4 activation because of the difference of GPCR sequences. Two conserved tyrosine residues (Y219[5.58] and Y302[7.53]) in TM5 and TM7 appeared to act as a hydrophobic switch. Upon activation, the two residues moved closer together, causing the breakage of the hydrophobic barrier. The distance measured between the oxygen atoms of the side chains of Y219[5.58] and Y302[7.53] was ~1.1 nm after 1.8 µs of simulation time, which was less than that at the initial time (~1.5 nm) for the CXCL12-bound CXCR4 system, whereas the distances measured in other CXCR4 systems after the simulation time ranged from 1.9 nm to 2.4 nm longer than the CXCL12-bound CXCR4 system (Fig. 3c). This finding indicates that the hydrophobic barrier is maintained to inhibit the internal water flowing. The conformation of CXCL12-bound CXCR4 system showed that Y302[7.53] flipped inward and TM7 tilted inward, whereas the conformation of CXCR4 in the inactive state for IT1t-bound CXCR4 system revealed that the two residues Y219[5.58] and Y302[7.53] did not flip inward, and TM7 tilted outward, which increased the distance (Supplementary Fig. S3g).

**Formation of internal water flow in receptor CXCR4 during activation**. Closer observation of the internal water distribution

from the final frame of each CXCR4 system revealed the presence of two HLs inside the TM region of CXCR4 (Fig. 4), which is similar to other GPCR systems[26,27]. For the CXCL12-bound CXCR4 system, water entered the receptor from both the EC and IC regions of the receptor, which is consistent with the observations of a previous computational study[27]. Internal water flow was concentrated near the EC region and hindered by the HL1 (F87[2.53], Y116[3.32], L120[3.36], and F292[7.43]), which enabled the internal water molecules to bypass HL1 and become trapped between Y256[6.52] and W252[6.48]. The TM7 moved inward and Y302[7.53] swung into the receptor interior, closer to Y219[5.58]. The water entered from the IC region and broke the HL2 (L80[2.46], L127[3.43], I130[3.46], and L244[6.40]), forming hydrogen bonding networks among the surrounding waters and residues through Y302[7.53] to reach the middle of CXCR4, but remained blocked by the HL1. The interaction of H294[7.45] with W252[6.48] can link the internal water pathway from both EC and IC regions during activation (Fig. 4a). The water entrance pathway computed by HOLE program[28] was also depicted in Supplementary Fig. S4a. In CXCL12-bound CXCR4 system, water molecules could easily enter from both EC and IC regions of CXCR4, mediated by W252[6.48] and H294[7.45]. However, two HLs appeared to hinder the internal water pathway in the apo and antagonist IT1t-bound CXCR4 systems, and several water molecules were trapped between HL1 and HL2 (Fig. 4b, c). In the IT1t-bound CXCR4 system, Y302[7.53] flipped down and did not swing to disrupt the HL2, which blocked the water entrance. A similar phenomenon to the apo CXCR4 system was observed in the

CXCL12-bound mCXCR4 system (Fig. 4d), whereby water molecules were trapped between HL1 and HL2, Y302$^{7.53}$ flipped up, and the mutant residue P244$^{6.40}$ moved inward to maintain the HL2, without forming a continuous water pathway.

The water density maps observed in the transmembrane region of respective simulation systems, analyzed from the simulation trajectories, revealed that HL1 was present in all systems, whereas HL2 was broken in the CXCL12-bound CXCR4 system to form an almost continuous water channel. Barely scattered waters in the inactive CXCR4 (apo, mCXCR4, IT1t-bound) systems were trapped between HL1 and HL2, and thus a continuous water pathway was not formed. (Supplementary Fig. S4b). The observed HL2 was also similar to previous studies in which a hydrophobic barrier was mentioned in the rhodopsin structure[25]. Sequence alignment of these residues from HL1 demonstrated that they were conserved among the Class A chemokine family of GPCRs (Tables 1 and 2). The HL1 residues Y116$^{3.32}$ and F292$^{7.43}$ flipped inside, adopting a closed conformation and acting as a barrier that caused the internal water to employ an alternative pathway in all simulation systems (Fig. 4). Other GPCR systems reported that HL1 and HL2 were broken to form a continuous water pathway when the receptor was activated[26,27], which differed from the CXCL12−CXCR4 system that water molecules bypass HL1, instead of breaking HL1. Nevertheless, these results indicated that HL1 may be present in all systems. The CXCL12-bound CXCR4 structure was more activated than the other three systems because of the breaking of HL2, which may form an approximately continuous water pathway and hydrogen bonding networks with surrounding residues for activation and downstream signal transmission. Moreover, as compared to previous comprehensive library mutations on CXCR4 receptor[9], some residues found in HL2 or regulating internal water flow were verified by mutagenesis experiments (Y219$^{5.58}$F, L244$^{6.40}$P, Y302$^{7.53}$H, W252$^{6.48}$R) to decline Ca$^{2+}$ mobilization as they were mutated. The formation of continuous internal water flow of GPCR may be crucial during GPCR activation.

**Interaction of the α5-helix of G$_{αi}$-protein with CXCL12-bound CXCR4 receptor**. Studies have suggested that the G$_{βγ}$ subunit of the G$_i$-protein facilitates the coupling of the G$_{αi}$ subunit to the receptor, and the GDP/GTP exchange in the G$_{αi}$ subunit engenders the dissociation and interaction with downstream effectors, such as G$_{αi}$ inhibition of adenylyl cyclase and G$_{βγ}$ activation of ion channels[29,30]. Furthermore, recently solved structure of rhodopsin-G$_i$ complex clearly revealed that the interactions between rhodopsin and G$_i$-protein are mainly mediated by the G$_{αi}$ subunit and somewhat by G$_{βγ}$ subunit[14,15,31]. Therefore, we only focused on a variation in the interaction pattern of G$_{αi}$-protein with CXCR4 receptor to reduce the computing consumption in our systems. To reveal the downstream signaling of G$_{αi}$-protein coupling with the activated CXCR4, we selected the last frame of 1.8-μs MD simulations of the CXCL12-bound CXCR4 system to couple with G$_{αi}$-protein by performing additional microsecond-scale MD simulations for the CXCL12–CXCR4–G$_{αi}$ tricomplex structure at various G$_{αi}$-protein states. The complex structure of nucleotide-bound G-protein binding to GPCR warrants clarification. Only less complex structures of nucleotide-free G-protein bound to GPCRs have been reported[14,15]. Therefore, three states of G$_{αi}$-protein coupling with the CXCL12-bound CXCR4 were simulated to clarify the interactions between G$_{αi}$-protein and CXCR4. The three states were GDP-bound G$_{αi}$-protein docked to CXCL12-bound CXCR4, nucleotide-free G$_{αi}$-protein coupled with CXCL12-bound CXCR4 through homology modeling using the μ-opioid receptor-G$_{αi}$ complex as the template[15], and GTP-bound G$_{αi}$-protein coupled

with CXCL12-bound CXCR4 through homology modeling. The docking results revealed that the α5-helix of GDP-bound G$_{αi}$-protein initially binds near the loop ICL3 of CXCR4 and the C-terminus of α5-helix facing the loop ICL2 of CXCR4. The α5-helix occupies the cytoplasmic space of CXCR4 among TM6, TM5, and TM3 (Supplementary Fig. S5). These findings all accord to those from previous GPCR-G$_{αi}$ complex structure studies[13–15].

MD simulations demonstrated that G$_{αi}$-protein undergoes various conformational changes when bound to GDP, GTP, and in the absence of the nucleotide. In the GDP-bound state, the α5-helix residues were attracted to the ICL2, TM3, and TM7 of the CXCL12-bound CXCR4 receptor. The surface charge distribution maps indicated that the binding interface of the Ras domain of G$_{αi}$-protein (more negatively charged) is attracted to the cytoplasmic region of CXCR4 (more positively charged) through electrostatic interactions (Supplementary Fig. S6a, b), which may trigger a counterclockwise rotation change from −8° to approximately −35° and an upper translation of approximately 4.0 Å of the α5-helix during MD simulations (Fig. 5a). In the nucleotide-free state, the α5-helix interaction changed to TM6, TM7, ICL1, and ICL3 of the CXCL12-bound CXCR4, causing a dissimilar clockwise rotation of ~68°, which is similar to the compared simulation of the solved rhodopsin (RHO)–G$_{αi}$-protein complex structure[14,15]. In the GTP bound to G$_{αi}$-protein state, the rotation of the α5-helix

**Table 1 Sequence alignment of Class A GPCRs: the chemokine family.**

| GPCRdb (Class A) | 2.53 × 53 | 3.32 × 32 | 3.36 × 36 | 7.43 × 42 |
|---|---|---|---|---|
| [Human] CCR1 | F | Y | L | Y |
| [Human] CCR2 | F | Y | Y | M |
| [Human] CCR5 | F | Y | F | M |
| [Human] CCR7 | F | Y | F | C |
| [Human] CXCR1 | F | K | F | F |
| **[Human] CXCR4** | **F** | **Y** | **L** | **F** |
| [Human] CX3CR1 | F | F | F | F |
| [Human] ACKR2 | F | Y | F | F |
| [Human] CCRL2 | F | Y | L | T |
| Consensus | F, 100% | Y, 78% | F, 56% | F, 44% |
| Hydrophobicity (%) | 100 | 89 | 100 | 89 |

The back bold text highlights the CXCR4 receptor.

**Table 2 Sequence alignment of Class A GPCR.**

| GPCRdb (Class A) | 2.53 × 53 | 3.32 × 32 | 3.36 × 36 | 7.43 × 42 |
|---|---|---|---|---|
| [Human] M3 receptor | I | D | S | Y |
| [Human] α2A-adrenoceptor | V | D | C | Y |
| [Human] β2-adrenoceptor | M | D | V | Y |
| [Human] D2 receptor | V | D | C | Y |
| [Human] D3 receptor | V | D | C | Y |
| **[Human] CXCR4** | **F** | **Y** | **L** | **F** |
| [Human] Rhodopsin | M | A | G | K |
| [Human] Opsin-3 | V | G | G | K |
| [Human] GPR1 | F | A | M | F |
| Consensus | V, 44% | D, 56% | C, 33% | Y, 56% |
| Hydrophobicity (%) | 100 | 33 | 67 | 78 |

The black bold text highlights the CXCR4 receptor.

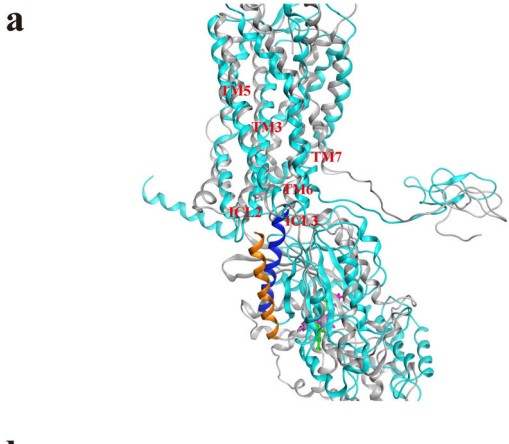

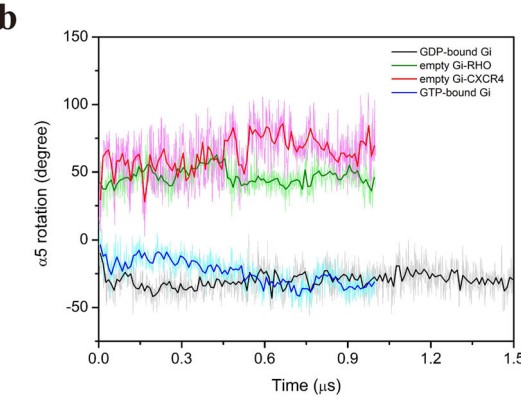

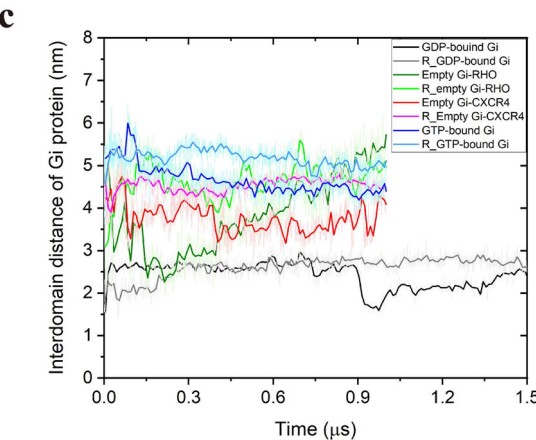

**Fig. 5 Conformational changes of $G_{\alpha i}$-protein induced by $G_{\alpha i}$-protein associated with the CXCL12-bound CXCR4 during the MD simulations.** **a** Superposition of CXCL12–CXCR4–GDP-bound $G_{\alpha i}$ tricomplex at the initial and final simulation phases. The complex structures are represented as ribbons, with gray color for initial time and cyan color for final time. For the initial time, α5-helix is colored orange, and GDP is represented as a green stick. For the final time, α5-helix is colored blue, and GDP is represented as a purple stick. **b** The rotation of the α5-helix of $G_{\alpha i}$-protein with time at various $G_{\alpha i}$-protein states. **c** The interdomain distance of $G_{\alpha i}$-protein with time at diverse $G_{\alpha i}$-protein states. For **b** and **c**, GDP-$G_{\alpha i}$-bound CXCR4, empty $G_{\alpha i}$-bound rhodopsin (RHO), empty (nucleotide-free) $G_{\alpha i}$-bound CXCR4, and GTP-$G_{\alpha i}$-bound CXCR4, are marked using black, green, red, and blue lines, respectively. The replicate simulations were shown as light colors.

changed to approximately −28° (Fig. 5b). Our simulations suggested that the rotation and translation of the α5-helix of $G_{\alpha i}$-protein is crucial in the nucleotide-exchange mechanism of $G_{\alpha i}$-protein bound to CXCR4.

**Interdomain distance of $G_{\alpha i}$-protein increases with GDP dissociation from the binding pocket.** During the simulation of CXCL12-bound CXCR4 in complex with GDP-bound $G_{\alpha i}$-protein, the Ras and the helical domains, which are initially tightly bound to the nucleotide, may gradually separate. The initial interdomain distance between A238 of the helical domain and E276 of the Ras domain was 15.6 Å. Over the course of the simulation, the distance rapidly increased and fluctuated around 29.5 Å before 750 ns and then decreased to 16.0 Å at 900 ns. The distance gradually increased again to 25.0 Å at the end of the simulation period (Fig. 5c), indicating that the separation of the helical domain from the Ras domain may exhibit spontaneous fluctuations similar to that of GDP-bound G-protein with receptor-free system[11]. In the nucleotide-free state, the distance between the two domains initially decreased before fluctuating around 40.0 Å, to maintain the opening of the two domains during the whole simulation. The compared simulation for the solved complex structure of empty (nucleotide-free) $G_{\alpha i}$-bound RHO receptor also maintained a similar opening profile, in which the interdomain distance fluctuated and increased during the simulation time. In the GTP bound to the empty $G_{\alpha i}$-protein state, the interdomain distance gradually decreased from 56.0 to 45.5 Å to stabilize GTP at the nucleotide-binding pocket (Fig. 5c). Moreover, during the simulation of GDP-bound $G_{\alpha i}$ system, the GDP was measured to be gradually released from the nucleotide-binding pocket (Supplementary Fig. S6c), which may be associated with the interdomain distance increase and fluctuation, and the rotation and uptranslation of the α5-helix upon $G_{\alpha i}$-protein activation. Although our current simulations could not present the GDP/GTP exchange during $G_i$-protein activation, we determined that the α5-helix rotation changed at different $G_{\alpha i}$ states, the interdomain distances slowly increased at GDP-bound $G_{\alpha i}$ state, and gradually decreased at GTP-bound $G_{\alpha i}$ state, which were crucial for $G_i$-protein activation[11,13–15].

**$G_{\alpha i}$-protein binding redirects the internal water flow in the CXCL12–CXCR4–$G_{\alpha i}$ tricomplex structure.** We finally examined the internal water flow in the CXCL12–CXCR4–$G_{\alpha i}$ tricomplex and further investigated the internal water distribution from the final frame of each CXCL12–CXCR4–$G_{\alpha i}$ tricomplex system. During the binding of GDP-bound $G_{\alpha i}$-protein, the nearly continuous water pathway was disturbed, with less water in the TM region of CXCR4 where HL2 reformed, thereby disrupting the water pathway (Fig. 6a). In the nucleotide-free $G_{\alpha i}$-protein state, $Y302^{7.53}$ swung into HL2 to break the hydrophobic gate and reformed an almost continuous internal water pathway (Fig. 6b). In the GTP-bound $G_{\alpha i}$-protein state, the continuous water molecules gradually decreased; the number of molecules was lower than that in the nucleotide-free state but higher than that in the GDP-bound $G_{\alpha i}$-protein state (Fig. 6c). The water density maps indicated that two HLs reformed to break the continuous water channel when GDP-bound $G_{\alpha i}$-protein coupled with CXCR4, whereas the continuous water flow reformed in the nucleotide-free $G_{\alpha i}$-protein state (Supplementary Fig. S4). This difference in the internal water pathway among $G_{\alpha i}$-protein coupling states may be associated with the $G_i$-protein activation.

**Molecular switches and HLs mediate internal water flow upon CXCR4 activation.** Internal water is essential to the function of numerous membrane proteins, such as channel and transport proteins, and continuous water flow has also been reported in activated GPCRs[27,32–35]. The presence of two HLs was proposed for the $A_{2A}R$ receptor, and during the active state of $A_{2A}R$, the NPxxY motif as a hydrophobic gate broke to enable water to flow through the TM of the receptor[27]. A similar mechanism was

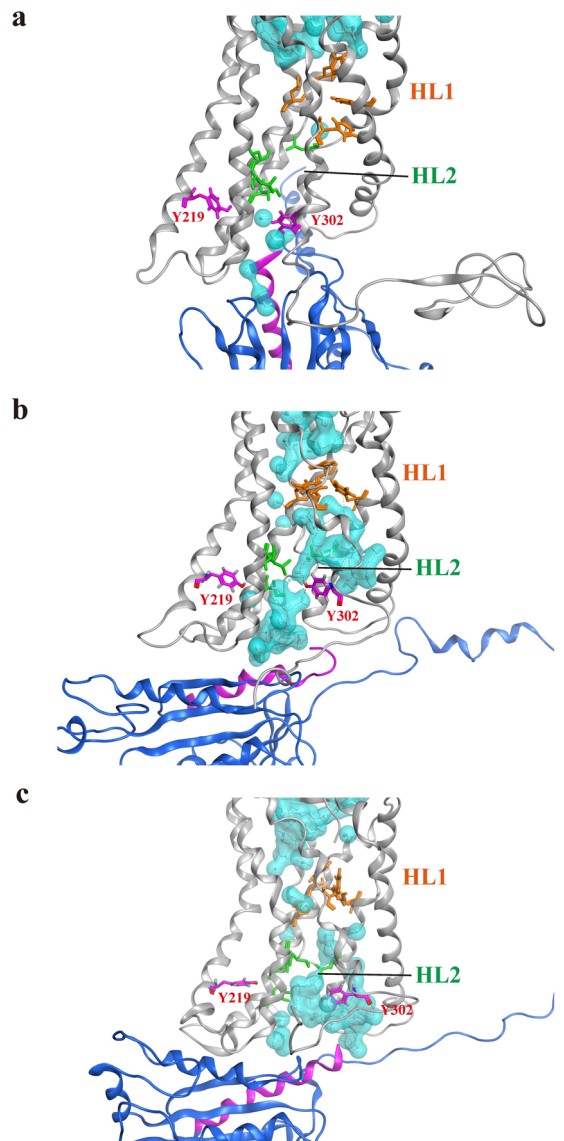

**Fig. 6 Internal water distribution within the transmembrane region of CXCR4 at various G_αi-bound states. a** GDP-bound G_αi protein.
**b** Nucleotide-free G_αi protein. **c** GTP-bound G_αi protein. Water molecules are drawn in both white sticks and in cyan for the molecular surface. The side chains of HL1 are marked as orange sticks, and the side chains of HL2 are marked as green sticks. CXCR4 is represented as a gray ribbon, and G_αi-protein is represented as a blue ribbon with α5-helix colored purple. Y302[7.53] and Y219[5.58] are depicted as purple sticks. Hydrogen bonding networks are also shown as red dotted line.

observed in our previous study, in which the HL broke during the active state of D3R receptor[26]. In the current study, we observed a relatively distinct water channel pathway, given that CXCR4 lacks ionic lock (distance between R3.50 and E6.30) and 3–7 lock (distance between E3.28 and K7.43) molecular switches which were proposed in previous experiments[25]. In the inactive states (apo, IT1t-bound CXCR4, and CXCL12-bound mCXCR4), the continuous water channel was hindered by the presence of two HLs. In the CXCL12-bound CXCR4 state, the HL2 broke because of the rearrangement of TM5 and TM7, and the distance between the two tyrosine residues Y219[5.58] and Y302[7.53] (tyrosine toggle switch) decreased (Figs. 4 and 3c). This observation is consistent with a previous study which reported that Y219[5.58] and Y302[7.53] underwent particularly large conformational changes from the

inactive to active state[9]. HL1 was retained throughout the simulation time because of the presence of hydrophobic residue clusters (F87[2.53], Y116[3.32], L120[3.36], and F292[7.43]) near the EC region to divert the internal water flow. Sequence alignment revealed that the presence of HL1 residues may be limited to only the chemokine family receptors (Tables 1 and 2); therefore, CXCR4 receptor had a highly divergent water diversion pathway when compared with A_2AR and D3R receptors. The hydrogen bonding network with time for various systems between TM residues and internal water molecules were calculated to reveal that more hydrogen bonding numbers were observed in the nucleotide-free G_αi state than in GDP-bound G_αi state, apo CXCR4 and IT1t-bound CXCR4 systems (Supplementary Fig. S7).

**Conformational changes of G_αi-protein cause GDP leaving induced by binding to CXCL12-bound CXCR4.** The α5-helix initially buried itself deep into the cytoplasmic cervices of CXCL12-bound CXCR4, and the α5-helix residues (I344, K345, and N347) in the GDP-bound state interacted with R148[ICL2], A303[7.54], A137[3.53], and K236[6.32] of the CXCR4 receptor. During the MD simulations, the α5-helix of G_αi-protein rotated counterclockwise and translocated closer to the cytoplasmic region of CXCR4 through electrostatic interactions (Supplementary Fig. S6a, b). In the meantime, the separation of the helical domain away from the Ras domain gradually increased the interdomain distance in the GDP-bound G_αi-protein state. Furthermore, the distances between switch I (C_α atom of T182) and switch II (C_α atom of G202) in the empty (nucleotide-free) G_αi state appeared larger than in the GDP-bound G_αi state (Fig. 7a). A free energy landscape as a function of the α5-helix and interdomain distance of G_αi-protein was performed to assess the stability of conformational states of tricomplex because these features are crucial in G-protein activation. The energy landscape for the α5-helix rotation and G_αi-protein interdomain distances clearly revealed three conformational states of CXCL12–CXCR4–G_αi tricomplex, GDP-bound G_αi, nucleotide-free G_αi, and GTP-bound G_αi states. The representative conformation of each state also demonstrated that more internal water was present in nucleotide-free G_αi state than in other states. (Fig. 7b). Moreover, the tyrosine toggle switch profiles with time for three different G_i-protein states indicated that the switch distance between Y219[5.58] and Y302[7.53] after G_i-protein binding was lower than in the inactive CXCR4 state (Supplementary Fig. S8). In summary, these results did not reveal the large domain opening of G_αi-protein for the GDP/GTP exchange, but did highlight trends in the rotation and uptranslation of the α5-helix and the increase in distance between switch I and switch II when the helical domain underwent a conformational change of G_αi-protein to increase the interdomain distance for GDP leaving from the binding pocket. These findings were supported by previous computational results[11,13] that the α5-helix interacts with the cytoplasmic pocket of the activated RHO receptor, causing displacement of the helical domain and GDP release. Recently solved structure of β_2AR in complex with 14 a.a. from C-terminal G_s with GDP-bound revealed that α5-helix rotation and tilt may differ depending on the G_αs-bound β_2AR states, which also corresponds to our simulation results[36].

**Internal water channel reformation for CXCL12-bound CXCR4 in complex with G_αi-protein.** Based the findings in the different simulation systems, an internal water channel formation model for CXCL12-bound CXCR4 in complex with G_αi-protein was proposed (Fig. 8). Because CXCL12 initially binds to CXCR4 through electrostatic interactions, K1 of the embedded N-terminus of CXCL12 interacted with E288[7.39] of CXCR4 to

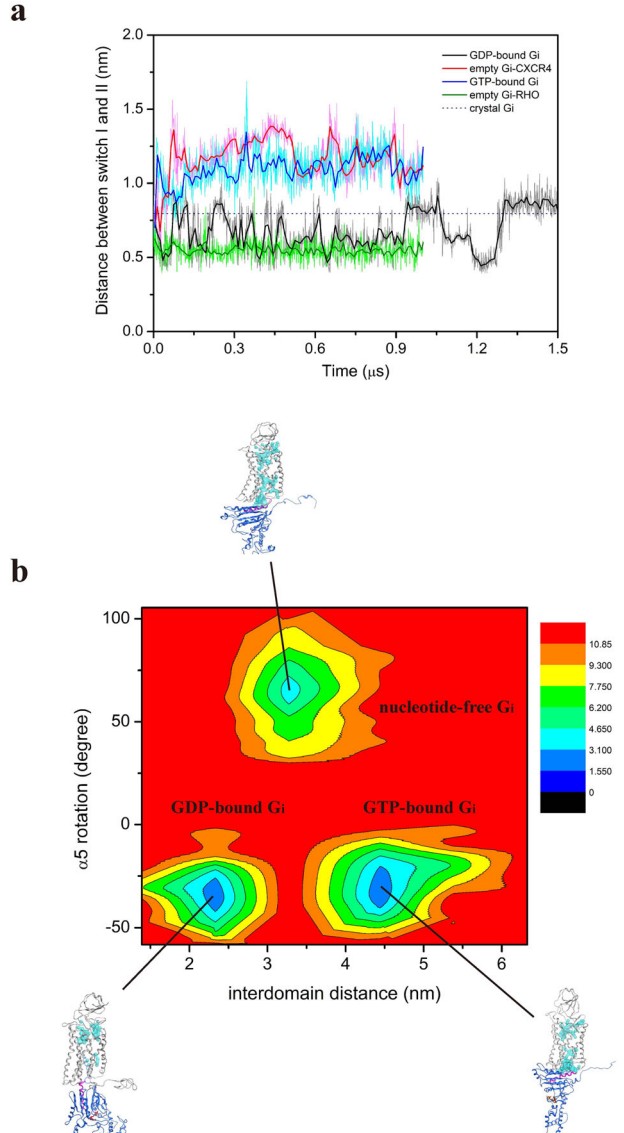

**Fig. 7 Molecular switches of $G_{\alpha i}$-protein and energy landscape at various $G_{\alpha i}$-proteins bound to pre-activated CXCR4 states. a** Distance between the $C_\alpha$ atoms of T182 of switch I and G202 of switch II of $G_{\alpha i}$ protein in diverse states. GDP-bound $G_{\alpha i}$, empty $G_{\alpha i}$-bound RHO, empty $G_{\alpha i}$-bound CXCR4, and GTP-bound $G_{\alpha i}$, are marked using black, green, red, and blue lines, respectively. The blue dash line represents the crystal structure of $G_{\alpha i}$ protein. In these simulations, the switch I and switch II distance increased in the empty state (11.5 Å) compared with the GDP-bound $G_{\alpha i}$ state (8.8 Å). The empty $G_{\alpha i}$ state is also called nucleotide-free $G_{\alpha i}$ state. **b** Free energy landscape of CXCL12-bound CXCR4 in complex with various $G_{\alpha i}$ states as a function of α5-helix rotation and interdomain distance. The energy unit used is kcal/mol.

activate the receptor and water flowing from the EC region was hindered by HL1, diverting the pathway, whereas the water from the IC region flowed through the broken HL2. During the activation, H294[7.45] interacted with W252[6.48] to link the water molecules from both the EC and IC regions, forming a nearly continuous water pathway. These two critical residues (W252[6.48] and H294[7.45]) have also been associated with CXCR4 activation in previous mutagenesis experiments[9,37–39]. TM6 moved outward for G-protein coupling, and TM7 moved inward to bring Y302[7.53] closer to Y219[5.58], disrupting the HL2. During the GDP-bound $G_i$-protein association with the cytoplasmic region of CXCR4, the

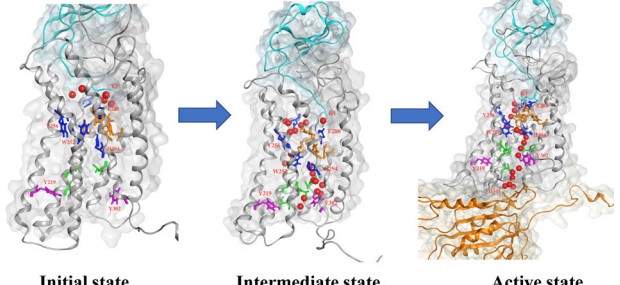

**Fig. 8 Internal water formation model for CXCL12-bound CXCR4 in complex with empty $G_{\alpha i}$-protein.** The complex structure is represented as a ribbon, with CXCL12 colored cyan, CXCR4 colored gray, and $G_{\alpha i}$-protein colored orange. Water molecules found in CXCR4 TM region during simulations are drawn as red balls. The side chains of HL1 are marked by orange sticks, and the side chains of HL2 are marked by green sticks. The main residues included in the internal water formation model are indicated by sticks. From left to right: initial state of agonist CXCL12 bound to receptor CXCR4; intermediate state of CXCL12 bound to CXCR4 after 1.8 μs MD simulations; active state of CXCL12-bound CXCR4 in complex with $G_{\alpha i}$-protein after 1.0 μs MD simulations.

continuous water pathway was disturbed and the α5-helix of $G_{\alpha i}$-protein was rotated and uptranslated to gradually enlarge the interdomain distance for GDP leaving. Then, Y302[7.53] swung to break HL2 and reform a continuous water pathway in the nucleotide-free $G_{\alpha i}$-protein state (Fig. 8). During GTP binding to nucleotide-free $G_{\alpha i}$-protein, α5-helix rotation decreased; this induced an interdomain distance decrease caused by $G_i$-protein conformational changes, which might have disrupted the continuous water pathway again (Fig. 7b).

**Conclusions.** In this study, the microsecond-scale MD simulations could not simulate the entire CXCR4 activation process. However, the atomic-level dynamic information allowed us to observe certain key intermolecular changes during CXCR4 activation, such as tyrosine toggle switch, HL breaks that form the continuous internal water pathway, and conformational differences of α5-helix of $G_{\alpha i}$-protein. The proposed internal water channel formation model for CXCL12-bound CXCR4 in complex with $G_{\alpha i}$-protein based on our MD results is valuable, and it might be useful for further understanding the activation mechanism of CXCR4 and in anticancer drug development.

## Methods

**Construction of receptor CXCR4 with N-terminus.** Because the solved crystal structure of CXCR4 (PDB: 3ODU) lacks the N-terminus[4], which is crucial in the initial binding of CXCL12 to CXCR4[6], a full-length CXCR4 was constructed by attaching a missing N-terminus (a.a.: 1–26) obtained from the Protein Data Bank (PDB: 2N55)[17] to form a fully modeled CXCR4 containing 327 residues (a.a.: 1–327) according to the method proposed by Lei Xu, et al[7]. First, we superposed the NMR structures of the CXCR4 N-terminus with the CXCR4 N-terminal segment in the crystal structure (residues 27–38). Then, the CXCR4 N-terminal segment (residues 27–38) in the NMR structure was deleted, and the best-oriented N-terminus (a.a.: 1–26) of CXCR4 from the NMR was attached manually to the crystal structure of CXCR4 using the Molecular Operating Environment (MOE) software package MOE2016.08 (http://www.chemcomp.com). The modeled CXCR4 receptor was then embedded into the 1-palmitoyl-2-oleoyl-sn-glycero-3-phosphocholine (POPC; 16:0−18:1 diester PC) lipid bilayer for energy minimization and equilibration. The equilibration procedure was described in the following "Molecular dynamics (MD) simulations" section.

**Molecular docking of chemokine CXCL12 to its receptor CXCR4.** The ligand–receptor docking between chemokine receptor CXCR4 and chemokine CXCL12 was conducted using the ZDOCK module of Discovery Studio 3.5 (BIO-VIA, http://accelrys.com). Previous studies have demonstrated that the N-terminal domain of receptor CXCR4 plays a crucial role in ligand CXCL12 binding[6,7,21]. A general two-sites model was also proposed that chemokine-receptor binding

**Table 3 Summary of simulation lengths and atom numbers.**

| Simulation system | Simulation time (µs) | Total number of atoms | Number of sodium ions | Number of chloride ions | Number of lipids |
|---|---|---|---|---|---|
| CXCL12–CXCR4 | $2 \times 1.8^a$ | 96,904 | 112 | 125 | 238 |
| apo CXCR4 | $2 \times 1.5$ | 86,191 | 101 | 105 | 238 |
| CXCL12-mCXCR4 | $2 \times 1.5$ | 85,329 | 100 | 113 | 238 |
| IT1t-CXCR4 | $2 \times 1.5$ | 70,107 | 83 | 86 | 238 |
| CXCL12-CXCR4-$G_{\alpha i}$-GDP | $2 \times 1.5$ | 105,212 | 118 | 122 | 242 |
| CXCL12-CXCR4-empty $G_{\alpha i}$ | $2 \times 1.0$ | 136,488 | 149 | 153 | 236 |
| CXCL12-CXCR4-$G_{\alpha i}$-GTP | $2 \times 1.0$ | 120,011 | 135 | 137 | 238 |
| Rhodopsin- $G_{\alpha i}$ | $2 \times 1.0$ | 115,385 | 135 | 130 | 242 |
| IT1t-CXCR4_crystal | 1.0 | 56,247 | 71 | 74 | 238 |
| CXCL12-CXCR4_Floudas's model | 1.0 | 85,452 | 100 | 113 | 244 |

$^a$Since the $G_{\alpha i}$ subunit is bound to the 1.8 µs CXCL12–CXCR4 complex, the different states of CXCR4-$G_{\alpha i}$ simulations are defined to start from 1.8 µs.

involves the interactions: between the N-loop of CXCL12 and the N-terminus of CXCR4 (site I), which was suggested as the interaction of the CXCL12 RFFESH loop with the N-terminal region of CXCR4, and between the N-terminal of CXCL12 and the extracellular region of CXCR4 (site II), which was reported as the N-terminus of CXCL12 embedded into the TM region of CXCR4[5–7]. Therefore, based on these experiments, we performed the docking of CXCL12 to receptor CXCR4. Initially, the chemokine CXCL12 was docked into receptor CXCR4, where the most part of TM region of the receptor and all the intracellular residues were blocked to allow the N-terminal domain, extracellular loops and several transmembrane residues for ligand binding. The ligand binding residues from S6-A21 were specified to interact with N-terminus of CXCR4 to enhance the docking accuracy[6,7,21]. ZDOCK searches conformational space by rotating the ligand around its geometric center with the receptor maintained fixed in space. The rotational search sampling grid was used as a 6° grid which sampled a total of 54,000 docked poses. The ZRANK function, part of the ZDOCK protocol, was used to rerank the top 2000 docked poses. Higher scores obtained from the ZDOCK program suggested that the complex structures were of higher quality. The poses generated from ZDOCK were clustered into a maximum of 50 groups. The RDOCK protocol was used for further refinement of the poses with higher ZDOCK scores, using a CHARMm-based energy minimization scheme for the optimization of intermolecular interactions. For more detailed settings of the ZDOCK module, please refer to our previous studies[20,40]. To determine the preferable docking poses, the lower RDOCK scores with lower binding energies, and the binding information from previous experiments were both evaluated. The structure with the lowest RDOCK scores was selected for further MD simulations.

**Molecular docking of antagonist IT1t to receptor CXCR4.** Antagonist IT1t was manually built using the MOE software package. The topology and parameter files of IT1t, not supported in the GROMACS program, were obtained from the GlycoBioChem PRODRG2 web server (http://davapc1.bioch.dundee.ac.uk/cgi-bin/prodrg) provided by Prof. Daan van Aalten[41] under the GROMOS 53A6 force field, which is also suitable for biomolecules. It was also confirmed using the Automated Topology Builder and Repository web server (ATB ver. 2.2)[42]. For more details on the docking of small compounds, please refer to our previous study[26].

**Molecular docking of GDP-bound $G_{\alpha i}$-protein to CXCL12-bound CXCR4.** To date, the complex structure of nucleotide-bound G protein binding to GPCR is still not available. To traverse the complete downstream signaling process of CXCL12-bound CXCR4 in complex with $G_{\alpha i}$-protein, the GDP-bound $G_{\alpha i}$ protein was docked to the cytoplasmic region of the pre-activated CXCR4. After carrying out 1.8-µs MD simulations for CXCL12-bound CXCR4, the final structure was selected to dock with GDP-bound $G_{\alpha i}$ using the ZDOCK module of Discovery Studio 3.5 (BIOVIA, http://accelrys.com), which the docking protocol is the same as CXCL12 docked to CXCR4. To increase the accuracy of docking, the TM and the extracellular regions of the receptor were blocked, and only the cytoplasmic region of CXCR4 was filtered.

**Homology modeling of CXCL12-bound CXCR4 in complex with $G_{\alpha i}$-protein.** The RHO–$G_{\alpha i}$ complex (PDB code: 6CMO)[14] and µ-OR–$G_{\alpha i}$ complex (PDB code: 6DDE)[15] were solved in 2018. To our knowledge, only the complex structures of GPCR-nucleotide-free G protein were solved, to obtain the preferable binding pose of CXCL12−CXCR4 bound to Gi-protein in the nucleotide-free and GTP-bound states, we followed the homology modeling method[13] by using the solved complex structure of the RHO–$G_{\alpha i}$ complex as a template to construct a comparative model for the CXCL12–$G_{\alpha i}$-bound CXCR4 tricomplex, such as CXCL12-nucleotide-free $G_{\alpha i}$-bound CXCR4 and CXCL12–$G_{\alpha i}$–GTP-bound CXCR4.

**Molecular dynamics (MD) simulations.** All MD simulation protocols were carried out according to our previous studies[20,26,40] using GROMOS 53A6 force field with the GROMACS 4.6.7 software package and an integration step size of 2 fs. All systems were embedded in the POPC lipid bilayer systems ($2 \times 144$ lipids), and the overlapping lipid was removed. The systems were hydrated using SPC216 water molecule. They were subsequently neutralized by adding ions ($Na^+$ and $Cl^-$) to generate 0.15 mol/L NaCl solution. The simulations were conducted in the NPT ensemble, employing a velocity-rescaling thermostat at the constant temperature of 310 K and 1 bar. Semi-isotropic pressure coupling was applied with a coupling time of 0.1 ps and a compressibility of $4.5 \times 10^{-5}$ $bar^{-1}$ for the $xy$-plane as well as for the $z$-direction. Long-range electrostatic interactions were calculated using the particle-mesh Ewald (PME) summation algorithm with grid dimensions of 0.12 nm and an interpolation order of 4. Lennard-Jones and short-range Coulomb interaction cut off values were 1.4 and 1.0 nm, respectively. The equilibration protocol was based on our previous studies[20,26,43] shown in the following, (i) the temperature was gradually increased from 100 K to 200 K and 310 K. The system was run for 500 ps for each temperature. During these simulations the complex structure remained fully restraint (k = 1000 kJ $mol^{-1}$ $nm^{-2}$). (ii) At 310 K the restraints kept on the complex structure via the force constant k, were released in 3 steps (k = 500, 250, 100 kJ $mol^{-1}$ $nm^{-2}$). Each step was run for 2.0 ns. After equilibration, production runs were carried out without any constraint in all structures. Details for all simulations were listed in Table 3. Two replicates were performed for each system with different initial random numbers to obtain similar results.

**Energy landscape calculations.** Previous studies have used free energy landscape (FEL) or potential of mean force (PMF) to assess the stability of protein conformations in local minimum energy state from numerous conformation changes[44,45]. Previous studies have suggested that the Ras domain α5-helix interacts with the cytoplasmic pocket of GPCR to trigger the displacement of the helical domain and GDP release. Upon activation of G protein coupling with GPCR, the interdomain distance may increase for GDP/GTP exchange[11,13,36]. In this study, we analyzed the distribution of the conformational states in terms of FEL or PMF as a function of α5-helix rotation and interdomain distance. The details were described in the following,

$$G(\alpha5, \text{interdomain}) = -k_B T lnP(\alpha5, \text{interdomain})$$

Where $P$, $T$, and $k_B$, are the probability distribution function, the absolute temperature, and the Boltzmann constant, respectively. The α5-helix rotation and interdomain distance with time were used to obtain the probability distribution $P$ ($\alpha5$, interdomain), which was computed based on the trajectory for each system at 310 K. GROMACS code g_sham was used to calculate the free energy landscape.

**MD simulation analysis.** VMD[46] with in-house scripts, GROMACS, and MOE software were used for visualization and analyses. The calculations of residue distance were performed by using GROMACS code g_bond. For tyrosine toggle switch, we measured the distance between the oxygen atoms of the side chains of Y219[5.58] and Y302[7.53] throughout MD simulation. For interdomain distance, the distance between $C_\alpha$ atoms of residues A238 and E276 of $G_{\alpha i}$-protein were measured. The hydrogen bonding network was calculated by using "Hydrogen Bonds" implemented in VMD for the MD trajectories of various systems. The water density maps were created by using GROMACS code g_densmap which analyzed the MD trajectories of various systems. The kink angle of TM6 with time for various simulation systems was calculated by using GROMACS code g_angle which three $C_\alpha$ atoms of residues (I245[6.41], P254[6.50], and G258[6.54]) were selected to measure the angle of TM6. The rotational angle of α5-helix of $G_{\alpha i}$-protein was calculated by using GROMACS code g_helixorient which calculated the coordinates and direction of the average axis inside the helix.

**Reporting summary**. Further information on research design is available in the Nature Research Reporting Summary linked to this article.

## Data availability

The data that support the findings of this study are available from the corresponding author upon reasonable request. The initial structure files and MD trajectories used in the study are deposited in persistent repository, figshare (https://figshare.com/s/8bd1a3615833d5b0e58f; https://figshare.com/s/b17197c734685c50dc01) (ref. [47]).

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

## Acknowledgements

H.J.H. acknowledges the Tzu Chi Medical Mission Project 106-03 (TCMMP 106-03) of the Buddhist Tzu Chi Medical Foundation for financial support. C.C.C. also acknowledges Hualien Tzu Chi Hospital Research Project (TCRD107-45) of the Buddhist Tzu Chi Medical Foundation for financial support. We acknowledge Taiwan National Center for High-Performance Computing for providing computer time and services. This paper was edited by Wallace Academic Editing.

## Author contributions

H.J.H., C.C.C., and J.W.L. conceived and designed the experiments. K.T.P.D., Y.T.L., S.F.P., and Y.C.Y. performed the computational simulations. H.J.H., K.T.P.D., C.C.C., S.J.J., and J.W.L. wrote the paper. All authors read and approved the final paper.

## Competing interests

The authors declare no competing interests.
