## [Peer Review File · Communications Chemistry]

Reviewers' comments:

Reviewer #1 (Remarks to the Author):

In the work, the author conducted multiple microsecond MD simulations to study the activation mechanism and internal water formation for CXCR4-CXCL12-Gai complex through analyzing the binding modes of CXCL12 and CXCR4, conformation changes of CXCR4 induced by CXCL12 and Gai protein induced by GDP leaving, internal water channel. The topic is important and interesting. Some meaningful results were obtained by the comparison of three different systems. However, some analyses are simple and some conclusions were not sufficiently supported by their data. Major revisions are required to improve the manuscript before acceptance. Some major concerns and few minor ones are shown in the following.

1. Why does the author only use the displacement of the residue 6.30 relative to the initial position? In addition, the author observed the TM6 displacement between the initial and the final frames, why doesn't the author further analyze the TM6 changes throughout the simulation?
2. It is also mentioned that previous studies already showed that G $\beta\gamma$ affects the coupling between Gai and receptor and the exchange of GDP/GTP. But, in the work, the authors only used Gai rather than the whole Gi protein to explore the interaction of G protein and CXCR4 and the mechanism of GDP release during G protein activation. More supports or descriptions should be given for rationalizing the usage of Gai.
3. How to select the representative conformations from the different states of energy landscape for the CXCR4-CXCL12-Gai complex? Why doesn't the author further analyze the structural features of these representative conformations and compare them? In fact, the authors compared structure differences based on the final frame, rather than the stable states of the Figure 7B, why?
4. It is unclear why the authors construct the mCXCR4 system (L2446.40P and L2466.42P). Have they been reported to have a significant effect on activation? Or you observed the effects of the mutations on receptor activation and internal water formation of CXCR4 through the work? The information should be given in the text.
5. In the part of "Molecular switches and HLs mediate the internal water flow upon CXCR4 activation", the author indicated that the HL2 is broke, judged from the distance between Y2195.58 and Y3027.53 in Figure 4. However, the changes of distance between the two residues are not obvious from Figure 4A to 4C. I suggest that they should monitor the change of distance between the two residues during the simulation. In addition, the distance between the two residues should be marked on the Figure 4.
6. Figure 4 is necessary to added more details. On line 250, it is mentioned that Y116 and F292 flipped inside and adopted a closed conformation, but the position of the residues is not indicated in the figure 4, so it is difficult to understand the discussion associated with the figure.
7. The model preparation should be described more clearly, for example, the building of CXCR4-CXCL12-Gai trimer.
8. How to calculate the rotation angle of $\alpha 5$ and why does the author select the $\alpha 5$ rotation and interdomain distance as reaction coordinates of FEL? These should be given in the text.
9. The repetition of some unimportant statements should be avoided.

Reviewer #2 (Remarks to the Author):

The manuscript authored by Chun-chun Chang et al describes a possible mechanism of ligand-induced activation of G-protein-coupled receptors (GPCR) and G-protein. They performed all-atom molecular dynamics simulation of GPCR (CXCR4) for microsecond-order, and analyzed conformation changes of the receptor upon ligand-binding, as well as water dynamics within the transmembrane regions. Although the results could not completely describe the activation mechanism of G-protein, they still contain several important implications. Those include i) a ligand-induced breakage of hydrophobic layer within the transmembrane region and the formation of a continuous water channel within it, ii) rotational and translational movement of α -5 helix

of GDP-bound G-protein upon docking to ligand-activated receptor, which induces a releasing motion of GDP, and iii) formation of a water channel within the transmembrane region in ligand-receptor-G-protein complex. I would like to support the publication of this manuscript in the journal. I only have several suggestions on the analysis of MD results.

The dynamics of water molecules inside the transmembrane region is very curious. The authors used the final frame of the MD trajectories to analyze water molecules (figure 3). However, it may be useful to investigate the following issues.

i) How waters entered into the channel from EC and IC regions. The authors mentioned H294 plays a role in guiding water from EC to the channel. It may be interesting to show the pathway of water into the channel.

ii) Rotational and translational freedom of water molecules at HL.

It can be speculated that the dynamics of water in the channel is restricted by neighboring residues. The dynamics of hydrogen bonding network may provide useful information on the characteristics of the channel.

Reviewer #3 (Remarks to the Author):

In this manuscript, the authors use classical all-atom molecular dynamics simulations to gain insights into the mechanism of activation of the chemokine receptor CXCR4 by its cognate chemokine agonist CXCL12, and how this translates into the activation of the Gi protein.

In their simulations, the authors observe conformational changes in the receptor and the G protein. Specifically, they report the disruption of a hydrophobic barrier in the receptor core that allows the formation of a continuous internal water channel. They also describe conformational changes in the G protein C-terminus and separation between the domains of the Gi alpha subunit, that they relate to GDP release.

These effects have been discussed several times in the literature, as the authors acknowledge in their references. For instance, the formation of a water channel between has been reported in e.g.:

- Nygaard, R., Valentin-Hansen, L., Mokrosinski, J., Frimurer, T. M., & Schwartz, T. W. (2010). Conserved water-mediated hydrogen bond network between TM-I, -II, -VI, and -VII in 7TM receptor activation. *The Journal of Biological Chemistry*, 285(25), 19625–36.

- Yuan, S., Filipek, S., Palczewski, K., & Vogel, H. (2014). Activation of G-protein-coupled receptors correlates with the formation of a continuous internal water pathway. *Nature Communications*, 5(1), 4733.

- Venkatakrisnan, A. J., Ma, A. K., Fonseca, R., Latorraca, N. R., Kelly, B., Betz, R. M., ... Dror, R. O. (2019). Diverse GPCRs exhibit conserved water networks for stabilization and activation. *Proceedings of the National Academy of Sciences*, 116(8), 201809251. Retrieved from <https://doi.org/10.1073/pnas.1809251116>

and others. Several of these published works already describe the 'tyrosine toggle switch' and the rearrangement of hydrophobic barriers in the receptor core discussed in this manuscript by the authors.

Similarly, the conformational changes in the G protein C-terminus and how they translate into GDP release have also been described in, e.g.:

- Rose, A. S., Zachariae, U., Grubmüller, H., Hofmann, K. P., Scheerer, P., & Hildebrand, P. W. (2015). Role of Structural Dynamics at the Receptor G Protein Interface for Signal Transduction. *PLOS ONE*, 10(11), e0143399.
- Dror, R. O., Mildorf, T. J., Hilger, D., Manglik, A., Borhani, D. W., Arlow, D. H., ... Shaw, D. E. (2015). Structural basis for nucleotide exchange in heterotrimeric G proteins. *Science (New York, N.Y.)*, 348(6241), 1361-5.
- Du, Y., Duc, N. M., Rasmussen, S. G. F., Hilger, D., Kubiak, X., Wang, L., ... Chung, K. Y. (2019). Assembly of a GPCR-G Protein Complex. *Cell*, 177(5), 1232-1242.e11.

Therefore, there is no much novelty in the author's findings. It is interesting that they may be able to point to specific residues in CXCR4 as responsible for these effects, but this will be only interesting to a narrow readership. The concept of 'water channels', 'opening of hydrophobic barriers', and 'rearrangement of tyrosines' is well known and studied.

More importantly, some parts of the text are difficult to interpret. The grammar should be revised, but also the reasoning of the authors is sometimes difficult to follow. Some statements and concepts are unclear, for instance:

- What is the significance of the 'tyrosine toggle switch'. This is not explained in the manuscript.
- The authors claim to have performed their simulations starting from a 'pre-activated' form of CXCR4. How is this state modeled? In which sense it is 'pre-activated'?
- Why did the authors simulate the L244(6.40)P and L246(6.42)P mutants? There is no explanation in the text. I presume L244(6.40)P was designed to disrupt one of the hydrophobic barriers and lead to a 'constitutively active' conformation, but this is never stated explicitly. And there is no mention on the role of L246(6.43). Also, why mutations to Pro? This mutation is likely to distort transmembrane helix 6. If there are experimental data showing that these mutations lead to constitutive activity (which I don't know), this likely involves other effects that disrupting the hydrophobic barrier. Still, none of this is addressed to in the manuscript.
- line 394: '... interactions between rhodopsin and Gi-protein are exclusively mediated by the Gai subunit'. This is not accurate; see, for instance:
- Tsai, C.-J., Marino, J., Adaixo, R., Pamula, F., Muehle, J., Maeda, S., ... Schertler, G. (2019). Cryo-EM structure of the rhodopsin-Gai-βγ complex reveals binding of the rhodopsin C-terminal tail to the Gβ subunit. *ELife*, 8, e46041.

This statement may be a justification for why the authors did not simulate the G protein ternary complex, but limited to Galpha. However, the authors do not provide any explicit explanation on why they made this choice. It is perfectly fine to use a simpler model system for the study, but this simplification needs to be justified to some extent. The authors do not provide any explanation on why they decided to exclude Gbeta (other than the statement in line 394, which is not accurate). Also, it does not seem that they 'anchored' Galpha to the membrane in their simulations. Was this a choice? If so, why?

- line 372: '... CXCR4 lacks ionic lock and 3-7 lock molecular switches.' The authors have to specify which ionic lock they are talking about. They seem to refer to the salt bridge between TM6 and TM3 present in some GPCRs but not in chemokine receptors; but the term 'ionic lock' commonly refers to an intra-helical salt bridge in TM3, which is indeed present in CXCR4. And what are the other 3-7 switches missing?

However, my main concerns are related to a general lack of details in the Methods section. Specifically:

- It is not clear how the modeling of the CXCR4 N-terminus was made.
- How was ICL3 of CXCR4 modeled? This region is not observed in the available crystal structures of CXCR4, and it is important for interactions with the G protein.
- The docking of CXCL12 into CXCR4 is not sufficiently described. The authors state that they initially placed CXCL12 based on some experimental data (line 490), but without providing any details. After using docking software, they only state that they used the solution with the lowest RDOCK score, as the 'scores obtained from the ZDOCK program suggested that the complex structures were of higher quality'. This is very ambiguous. In the context of this study, the binding mode of the agonist is an important point, and the authors do not describe in sufficient detail what were the criteria that they used in their docking experiments.
- Also, why didn't the authors use any of the experimental structures of chemokines bound to their receptors in their docking? This seems like a valuable piece of information to guide the docking of CXCL12 into CXCR4, but looks like the authors didn't consider it. And why did they re-dock the small-molecule antagonist IT1t to CXCR4, if there is already a crystal structure available?
- The authors claim that their microsecond-scale simulations might have allowed CXCR4 to reach an intermediate state. While it is difficult to judge what the authors exactly observed without analyzing the trajectories, the kinetics of CXCR4 activation are several orders of magnitude slower. Specifically, FRET measurements have revealed that CXCL12 binding to CXCR4 results in structural rearrangements within the transmembrane domains of the receptor in approx. 600 ms, of rearrangements between CXCR4 and the G protein in approx. 1 second and G protein activation in approx. 4 seconds.
- Perpina-Viciano, C., Isbilir, A., Zarca, A., Caspar, B., Kilpatrick, L. E., Hill, S. J., ... Hoffmann, C. (2020). Kinetic analysis of the early signaling steps of the human chemokine receptor CXCR4. *Molecular Pharmacology*. DOI: 10.1124/mol.119.118448.

The authors should be more cautious about suggesting that their microsecond-scale simulations allow the receptor to reach intermediate active states.

While the conclusions of this work seem reasonable –albeit already established in several other GPCRs– I don't think the data presented in the manuscript fully support them. The authors performed medium-length simulations in simplified systems (without providing enough justification on their choices) and they probably observed some rearrangements, as expected in a molecular dynamics simulation. Then, it seems that they simply assigned these trends to already observed phenomena. I can't see a well-thought justification on how the authors reach their conclusions from their simulations.

In summary, I think the authors should heavily revise this manuscript and perhaps submit to a more specialized journal.

Reply to Reviewer 1

The authors thank reviewer for the very helpful comments. Replies to the comments are given in the following. Also, the changes made in the manuscript are highlighted in yellow in the revised manuscript.

Reviewer #1 (Remarks to the Author):

1. Why does the author only use the displacement of the residue 6.30 relative to the initial position? In addition, the author observed the TM6 displacement between the initial and the final frames, why doesn't the author further analyze the TM6 changes throughout the simulation?

[Response]

The authors thank for reviewer's suggestion. As mentioned in the manuscript, during the GPCR activation, the down-half region of TM6 shows larger outward movement; therefore, we selected the last residue of TM6, K6.30 to measure its displacement between different time frames, and superposed the CXCR4 structures at different time frames to show TM6 movement. The measurement and superposition were also shown in other GPCR papers (ref. 10,30 in the manuscript). According to reviewer's suggestion, we also add another analysis of TM6 changes (TM6 kink angle) with time in the revised manuscript.

2. It is also mentioned that previous studies already showed that $G\beta\gamma$ affects the coupling between $G\alpha_i$ and receptor and the exchange of GDP/GTP. But, in the work, the authors only used $G\alpha_i$ rather than the whole Gi protein to explore the interaction of G protein and CXCR4 and the mechanism of GDP release during G protein activation. More supports or descriptions should be given for rationalizing the usage of $G\alpha_i$.

[Response]

The authors thank for reviewer's suggestion. As chemokine CXCL12 binds to CXCR4, the conformational changes occur in the TM and IC regions of CXCR4, acting as signals for the heterotrimeric inhibitory G-protein (G_i) binding. Once the trimeric G_i -protein couples with the active receptor, the $G_{\alpha i}$ subunit undergoes a conformational change, which promotes the domains separation in $G_{\alpha i}$ subunit and the exchange of GDP to GTP for downstream signaling. Previous studies have indicated that $G_{\beta\gamma}$ subunit of G_i -protein facilitates the coupling of $G_{\alpha i}$ subunit and the GDP/GTP exchange in the $G_{\alpha i}$ subunit leads to the dissociation and interaction with downstream effectors such as $G_{\alpha i}$ inhibition of adenylyl cyclase and $G_{\beta\gamma}$ activation of ion channels (ref. 29,30 in the manuscript). In addition, recently solved structure of rhodopsin- G_i complex clearly revealed that the interactions between rhodopsin and G_i -protein are mainly mediated by the $G_{\alpha i}$ subunit (ref. 14,15,31 in the manuscript). Therefore, to reduce the computing consumption in our systems, in this study, we only focused on a variation in the interaction pattern of $G_{\alpha i}$ -protein with CXCR4 receptor. In the simulation of CXCL12-CXCR4- $G_{\alpha i}$ tricomplex structure, the $\alpha 5$ -helix of $G_{\alpha i}$ rotation clockwise and up-translation to the cytoplasmic region of CXCR4 may cause the conformational changes of $G_{\alpha i}$ subunit, such as the distance increase between the switch I (T182) and switch II (G202) of $G_{\alpha i}$ subunit, and gradual increase of domains separation of $G_{\alpha i}$ to trigger GDP release. We add some sentences in the revised manuscript.

3. How to select the representative conformations from the different sates of energy landscape for the CXCR4-CXCL12- $G_{\alpha i}$ complex? Why doesn't the author further analyze the structural features of these representative conformations and compare them? In fact, the authors compared structure differences based on the final frame, rather than the stable states of the Figure 7B, why?

[Response]

As mentioned in "Materials and Methods" section, previous studies have used free energy landscape to assess the stability of protein conformations in local minimum energy state from numerous conformation changes. In this study, we analyzed the distribution of the conformational states in terms of FEL as a function of $\alpha 5$ -helix rotation and interdomain distance. We selected the local energy minimum as the representative conformation in the three $G_{\alpha i}$ -protein-bound CXCR4 states, which also meant higher probability distribution. The conformations shown in Figure 7B are different from those shown in Figure 6 which are the final frame of each CXCL12-CXCR4- $G_{\alpha i}$ tricomplex system. The structural comparison of these three states, such

as α 5-helix rotation, interdomain distance, switch I/II distance of $G_{\alpha i}$ protein over time were also shown in the manuscript. We add some sentences in the revised manuscript (highlighted in yellow).

4. It is unclear why the authors construct the mCXCR4 system (L2446.40P and L2466.42P). Have they been reported to have a significant effect on activation? Or you observed the effects of the mutations on receptor activation and internal water formation of CXCR4 through the work? The information should be given in the text.

[Response]

Previous mutagenesis experiments indicated that 6.40 and 6.44 positions in different receptors support a role in mediating the transition between inactive and active GPCR states (ref. 9,16 in the manuscript). In addition, two residues (L244^{6.40}P and L246^{6.42}P) were also deemed critical in CXCR4 signaling, which suggested Proline mutation in the region can eliminate signaling without altering extracellular structure or ligand binding (ref. 4,9,21 in the manuscript). In our study, for CXCL12-bound mutant CXCR4 system, the intercellular half region of TM6 tilted inward to shrink the cytoplasmic binding region for G_i protein and blocked the internal water flow when compared with CXCL12-bound CXCR4 system. The simulation results provided the atomic level insight and were consistent with these previous experiments and suggested that the conformation of intercellular half region of TM6 is important for signal transmission. We add detailed explanation in the revised manuscript.

5. In the part of “Molecular switches and HLs mediate the internal water flow upon CXCR4 activation”, the author indicated that the HL2 is broke, judged from the distance between Y2195.58 and Y3027.53 in Figure 4. However, the changes of distance between the two residues are not obvious from Figure 4A to 4C. I suggest that they should monitor the change of distance between the two residues during the simulation. In addition, the distance between the two residues should be marked on the Figure 4.

[Response]

The authors thank for reviewer’s suggestion. As mentioned in the manuscript, two conserved tyrosine residues (Y219^{5.58} and Y302^{7.53}) in TM5 and TM7 appeared to act as a hydrophobic switch (also called tyrosine toggle switch). Upon activation, the two residues came closer, causing the breakage of the hydrophobic barrier; therefore, the distance measured between the oxygen atoms of the side chains of Y219^{5.58} and Y302^{7.53} with simulation time for various systems was shown in Figure 3C of the manuscript. We add the mark of the distance between the two residues in Figure 4.

6. Figure 4 is necessary to added more details. On line 250, it is mentioned that Y116 and F292 flipped inside and adopted a closed conformation, but the position of the residues is not indicated in the figure 4, so it is difficult to understand the discussion associated with the figure.

[Response]

The authors thank for reviewer's suggestion. We add the labels of residues Y116 and F292 in the Figure 4 of the revised manuscript.

7. The model preparation should be described more clearly, for example, the building of CXCR4-CXCL12-G α i trimer.

[Response]

To study the interactions between CXCL12-bound CXCR4 and G α i protein at different states, we have to prepare the CXCL12–CXCR4–G α i tricomplex based on molecular docking and homology modeling methods. To date, the complex structure of nucleotide-bound G protein binding to GPCR is still not available. Due to the lack of the complex of GPCR–G α i with GDP bound and α 5-helix plays a crucial role in the G-protein activation, the GDP-bound G α i protein was docked to the cytoplasmic region of the CXCL12-bound CXCR4. After carrying out 1.8- μ s MD simulations for CXCL12-bound CXCR4 system, the final structure was selected to dock with GDP-bound G α i using the ZDOCK module of Discovery Studio 3.5, which the docking protocol is the same as CXCL12 docked to CXCR4. To increase the accuracy of docking, the TM and the extracellular regions of the receptor were blocked, and only the cytoplasmic region of CXCR4 was filtered. The preferable docking pose with lowest RDOCK score was selected for further MD simulations. To date, only the complex structures of GPCR–nucleotide free G protein were solved; therefore, we utilized the homology modeling method of MOE software by using the solved structure of the Rhodopsin–G α i complex as a template to construct these models, CXCL12–nucleotide-free G α i-bound CXCR4 and CXCL12–G α i–GTP-bound CXCR4. More detailed description is added in the revised manuscript.

8. How to calculate the rotation angle of α 5 and why does the author select the α 5 rotation and interdomain distance as reaction coordinates of FEL? These should be given in the text.

[Response]

The authors thank for reviewer's comments. The rotation angle of α 5-helix of G α i protein was calculated by using Gromacs code *g_helixorient* which calculates the coordinates and direction of the average axis inside the helix.

Previous studies have suggested that the Ras domain $\alpha 5$ -helix interacts with the cytoplasmic pocket of GPCR to trigger the displacement of the helical domain and GDP release (ref. 11,13,36 in the manuscript). Upon activation of G protein coupling with GPCR, the interdomain distance may increase for GDP/GTP exchange. Therefore, we selected these two important features as the reaction coordinates of FEL. More detailed explanation of FEL is added in the revised manuscript.

9. The repetition of some unimportant statements should be avoided.

[Response]

The authors thank for reviewer's comments. We have removed the repetition of some unimportant statements, shown in the revised manuscript.

Reply to Reviewer 2

The authors thank reviewer for the very helpful comments. Replies to the comments are given in the following. Also, the changes made in the manuscript are highlighted in yellow in the revised manuscript.

Reviewer #2 (Remarks to the Author):

The dynamics of water molecules inside the transmembrane region is very curious. The authors used the final frame of the MD trajectories to analyze water molecules (figure 3). However, it may be useful to investigate the following issues.

i) How waters entered into the channel from EC and IC regions. The authors mentioned H294 plays a role in guiding water from EC to the channel. It may be interesting to show the pathway of water into the channel.

[Response]

The authors thank for reviewer's suggestion. As mentioned in the manuscript, according to previous simulation study by Yuan et al. (ref. 27 in the manuscript) that the smaller NELs (N-terminal and ECL2 loops) would leave the orthosteric site widely open for waters entrance. In our simulations, we also found waters entered the receptor from both EC and IC regions during activation in CXCL12-CXCR4 system, shown in the following figure.

The analysis was performed by “HOLE” program, which showed that waters could easily enter the receptor from EC and IC regions (blue color), and HL2 was broken to allow waters passing through (green color). H294 interacted with W252 to link the continuous water pathway, which was also consistent with water density maps ([Figure](#)

S4B) obtained by analyzing the MD trajectories. Detailed description is added in the revised manuscript.

ii) Rotational and translational freedom of water molecules at HL.

It can be speculated that the dynamics of water in the channel is restricted by neighboring residues. The dynamics of hydrogen bonding network may provide useful information on the characteristics of the channel.

The authors thank for reviewer's suggestion. We add the calculation of hydrogen bonding network of waters inside the receptor through the simulation time. The result is consistent with our water density maps shown in Figure S4B. Detailed description is added in the revised manuscript.

Reply to Reviewer 3

The authors thank reviewer for the very helpful comments. Replies to the comments are given in the following. Also, the changes made in the manuscript are highlighted in yellow in the revised manuscript.

Reviewer #3 (Remarks to the Author):

.....

Therefore, there is no much novelty in the author's findings. It is interesting that they may be able to point to specific residues in CXCR4 as responsible for these effects, but this will be only interesting to a narrow readership. The concept of 'water channels', 'opening of hydrophobic barriers', and 'rearrangement of tyrosines' is well known and studied.

More importantly, some parts of the text are difficult to interpret. The grammar should be revised, but also the reasoning of the authors is sometimes difficult to follow.

[Response]

The authors thank for reviewer's comments. The binding and activation mechanisms of GPCRs are crucial, and many scientists have put many efforts on them. As the reviewer mentioned, in the past decades, although some GPCR structures were solved and several concepts were proposed and verified, there are still some questions needed to be answered. There is no much novelty in our research findings; however, in our studies, we proposed the importance of the formation of internal water channel and internal molecular switch of CXCL12-CXCR4-Gi complex, which is seldom mentioned. Moreover, CXCR4 is known to be a prognostic marker which is suggested to be associated with many cancers, such as breast, lung, and colon cancers, where it promotes metastasis, angiogenesis, and tumor growth or survival; therefore, our results may provide valuable atomic-level dynamic information in the development of anticancer and antimetastatic drugs.

The manuscript was sent for professional English editing. The English editing certificate is also attached. The manuscript is considered to be improved in grammar, punctuation, spelling, verb usage, sentence structure, conciseness, and general readability.

-What is the significance of the ‘tyrosine toggle switch’. This is not explained in the manuscript.

[Response]

It is well known that GPCRs are often depicted as molecular machines that alternate between the inactive and active states through the conversion of molecular switches. As mentioned in our previous study (ref. 26 in the manuscript) that continuous water pathways are important mediators of active GPCRs; however, the hydrophobic layers within the transmembrane region hinder the formation of continuous waters pathways. Once activated, the rotation of TM6 allows a hydrophobic layer to open, Tyr5.58 and Tyr7.53 rearrange their side chains to fill this gap. The movement of these two residue side chains allows additional interactions with waters ending hydrogen bond networks towards the DRY motif, resulting in the disruption of the ionic lock salt bridge. This molecular switch was called “tyrosine toggle switch”, explained by Standfuss et al. (Nature 471, 656-660, 2011) We add the explanation of “tyrosine toggle switch” in the revised manuscript.

- The authors claim to have performed their simulations starting from a ‘pre-activated’ form of CXCR4. How is this state modeled? In which sense it is ‘pre-activated’?

[Response]

The authors thank for reviewer’s comments. As mentioned in our manuscript, although the microsecond-scale MD simulations were performed for CXCL12 binding to CXCR4, it might just reach some state toward activation. In our manuscript, we called the intermediate state toward activation as pre-active state. To reveal the downstream signaling of G_{ai}-protein coupling with the activated CXCR4, we selected

the last frame of 1.8- μ s MD simulations of the CXCL12-bound CXCR4 system as the pre-active state to couple with G protein to perform additional microsecond-scale MD simulations for the CXCL12–CXCR4–G_{ai} tricomplex structure at various G_{ai}-protein states. Due to much longer time required to reach the real GPCR active state, to avoid confusion we correct the description of “pre-active” state in the revised manuscript.

- Why did the authors simulate the L244(6.40)P and L246(6.42)P mutants? There is no explanation in the text. I presume L244(6.40)P was designed to disrupt one of the hydrophobic barriers and lead to a ‘constitutively active’ conformation, but this is never stated explicitly. And there is no mention on the role of L246(6.43). Also, why mutations to Pro? This mutation is likely to distort transmembrane helix 6. If there are experimental data showing that these mutations lead to constitutive activity (which I don’t know), this likely involves other effects that disrupting the hydrophobic barrier. Still, none of this is addressed to in the manuscript.

[Response]

Previous mutagenesis experiments indicated that 6.40 and 6.44 positions in different receptors support a role in mediating the transition between inactive and active GPCR states (ref. 9,21 in the manuscript). In addition, two residues (L244^{6.40}P and L246^{6.42}P) were also deemed critical in CXCR4 signaling, which suggested Proline mutation in the region can eliminate signaling without altering extracellular structure or ligand binding (ref. 4,9,21 in the manuscript). In our study, to compare with these mutational experiments, we built the mutant CXCR4 with the two residues mutated as Proline (L244^{6.40}P and L246^{6.42}P). For CXCL12-bound mutant CXCR4 system, the intercellular half region of TM6 tilted inward to shrink the cytoplasmic binding region for G_i protein and blocked the internal water flow when compared to CXCL12-bound CXCR4 system. The simulation results provided the atomic-level insight and were consistent with these previous experiments and suggested that the conformation of intercellular half region of TM6 is important for signal transmission. We add more explanation in the revised manuscript.

- line 394: ‘... interactions between rhodopsin and Gi-protein are exclusively mediated by the G_{ai} subunit’. This is not accurate; see, for instance:
- Tsai, C.-J., Marino, J., Adaixo, R., Pamula, F., Muehle, J., Maeda, S., ... Schertler, G. (2019). Cryo-EM structure of the rhodopsin-G_{ai}- $\beta\gamma$ complex reveals binding of the rhodopsin C-terminal tail to the G β subunit. *ELife*, 8, e46041.

This statement may be a justification for why the authors did not simulate the G protein ternary complex, but limited to Galpha. However, the authors do not provide any explicit explanation on why they made this choice. It is perfectly fine to use a simpler model system for the study, but this simplification needs to be justified to some extent. The authors do not provide any explanation on why they decided to exclude Gbeta (other than the statement in line 394, which is not accurate). Also, it does not seem that they ‘anchored’ Galpha to the membrane in their simulations. Was this a choice? If so, why?

[Response]

The authors thank for reviewer’s suggestion. As mentioned in the manuscript, recently solved structure of rhodopsin-G_i complex by Kang et al. (Nature 558: 553-558, 2018) clearly revealed that the interactions between rhodopsin and G_i-protein are exclusively mediated by the G_{αi} subunit, and that there is no contact between rhodopsin and Gβγ subunit. However, as mentioned by the reviewer, the complex structure solved by Tsai et al. (Elife 8, 2019) showed that the C-terminal tail of rhodopsin binds to Gβ subunit, which was not found in the complex structure solved by Kang et al. These GPCR-G protein complex structures showed that Gα subunit is crucial for the binding of G protein to GPCR. Therefore, based on these reasons and to reduce the computing consumption for the whole CXCL12-CXCR4-G_iαβγ complex system, we only chose the Gα to bind to CXCR4 for our tricomplex simulations. In our CXCL12-CXCR4-Gα complex simulations, we did not anchor Gα subunit to the membrane by covalent bonding, but the long N-terminus of Gα subunit was a little bit embedded into the membrane, because Gα subunit would leave the receptor after GDP/GTP exchange in the Gα subunit as G protein activation. We correct the statement about Gβγ subunit and add the explanation for only using Gα subunit in the revised manuscript.

- line 372: ‘... CXCR4 lacks ionic lock and 3–7 lock molecular switches.’ The authors have to specify which ionic lock they are talking about. They seem to refer to the salt bridge between TM6 and TM3 present in some GPCRs but not in chemokine receptors; but the term ‘ionic lock’ commonly refers to an intra-helical salt bridge in TM3, which is indeed present in CXCR4. And what are the other 3-7 switches missing?

[Response]

From the first obtained GPCR X-ray structure, the inactive state of bovine rhodopsin, an intact salt bridge between R3.50 of the DRY motif in TM3 and E6.30 in TM6 was observed. In their work, it was hypothesized that the interaction between these two residues could be a key factor that constrains the receptor in its inactive form

(Palczewski, K. et al., *Science* 289, 739–745 (2000)). Many studies also refer the salt bridge between TM3 and TM6 as “ionic lock”. CXCR4 is lacking the residue E6.30 replaced by K6.30 and that no ionic lock is present between TM3 and TM6. The 3-7 lock switch was first observed in rhodopsin, where the salt bridge between E113^{3,28} and K296^{7,43} was confirmed to stabilize the receptor in its inactive conformation, and to be disrupted upon activation (Kim, J. M. et al., *PNAS*, 12508–12513 (2004)). In CXCR4, the 3-7 lock switch is also lacking, which the two residues are V112^{3,28} and F292^{7,43} instead of E113^{3,28} and K296^{7,43}. The detailed descriptions about ionic lock and 3-7 lock are added in the revised manuscript.

However, my main concerns are related to a general lack of details in the Methods section. Specifically:

- It is not clear how the modeling of the CXCR4 N-terminus was made.

[Response]

The authors thank for reviewer’s comments. As mentioned in the manuscript, due to the lack of the N-terminus of solved CXCR4 structure, which is crucial in the initial binding of CXCL12 to CXCR4, we followed the method proposed by Lei Xu *et al.* (ref. 7 in the manuscript) to construct a CXCR4 structure with N-terminus. First, we superposed the NMR structures of the CXCR4 N-terminus with the CXCR4 N-terminal segment in the crystal structure (residues 27 to 38). Then, the CXCR4 N-terminal segment (residues 27 to 38) in the NMR structure was deleted, and the best-oriented N-terminus (a.a.: 1–26) of CXCR4 from the NMR was attached manually to the crystal structure of CXCR4 using the Molecular Operating Environment (MOE) software package MOE2016.08. The modeled CXCR4 receptor was then embedded into a 1-palmitoyl-2-oleoyl-sn-glycero-3-phosphocholine (POPC; 16:0–18:1 diester PC) lipid bilayer for energy minimization and equilibration. More detailed description of modeling the N-terminus of CXCR4 is added in the revised manuscript.

- How was ICL3 of CXCR4 modeled? This region is not observed in the available crystal structures of CXCR4, and it is important for interactions with the G protein.

[Response]

Most GPCR structures have long intracellular loop 3 (ICL3), which may be important for the interactions with G protein, but the situation is different in chemokine receptors which ICL3 is quite short (~4 a.a.). The solved crystal structure of CXCR4 (PDB: 3ODU) contains a T4 lysozyme (T4L) fusion inserted between TM5 and TM6 at the cytoplasmic side of the receptor. The ICL3 of CXCR4 is only 4 residues

(KGHQ) available in the solved crystal structure. In our simulation systems, we removed T4L from CXCR4 structure and manually connected TM5 and ICL3-TM6 by using MOE molecular simulation software, and all of our CXCR4 systems contained ICL3.

- The docking of CXCL12 into CXCR4 is not sufficiently described. The authors state that they initially placed CXCL12 based on some experimental data (line 490), but without providing any details. After using docking software, they only state that they used the solution with the lowest RDOCK score, as the ‘scores obtained from the ZDOCK program suggested that the complex structures were of higher quality’. This is very ambiguous. In the context of this study, the binding mode of the agonist is an important point, and the authors do not describe in sufficient detail what were the criteria that they used in their docking experiments.

[Response]

The authors thank for reviewer’s comments. As mentioned in the manuscript, previous experiments have demonstrated that the N-terminal domain of receptor CXCR4 plays a crucial role in ligand CXCL12 binding. A general two-sites model was also proposed that chemokine-receptor binding involves the interactions: between the N-loop of CXCL12 and the N-terminus of CXCR4 (site I), which was suggested as the interaction of the CXCL12 RFFESH loop with the N-terminal region of CXCR4, and between the N-terminal of CXCL12 and the extracellular region of CXCR4 (site II), which was reported as the N-terminus of CXCL12 embedded into the TM region of CXCR4 (ref. 5-7 in the manuscript). Therefore, based on these experiments, we performed the docking of CXCL12 to receptor CXCR4. Initially, the chemokine CXCL12 was docked into receptor CXCR4, where the most part of TM region of the receptor and all the intracellular residues were blocked to allow the N-terminal domain, extracellular loops and several transmembrane residues for ligand binding. Before using ZDOCK module for CXCL12 binding to CXCR4, to increase the accuracy of docking, the TM region of the receptor was blocked and the ligand binding residues from S6-A21 were specified according to previous experiments (ref. 6,7,20 in the manuscript). For ZDOCK program, the rotational search sampling grid is used as a 6° grid which samples a total of 54000 docked poses. ZDOCK searches conformational space by rotating the ligand around its geometric center with the receptor kept fixed in space. Higher scores obtained from the ZDOCK program suggested that the complex structures were of higher quality. The poses generated from ZDOCK were clustered into a maximum of 50 groups. Then, the ZRANK function was used to rerank the top 2000 docked poses. Finally, the RDOCK protocol

was used for further refinement by using a CHARMM-based energy minimization scheme for the optimization of intermolecular interactions. For more detailed settings of the ZDOCK module, please refer to our previous studies (ref. 20, 26 in the manuscript). To determine the preferable docking poses, lower RDOCK scores with lower binding energies, and binding information from previous experiments were both evaluated. For CXCL12 docking to CXCR4 system, although the difference of RDOCK energy between different docking poses was not quite large, the binding conformations were noticeably different, and only the docking pose1 showed the conformation of CXCL12 embedded into CXCR4 (Figure S1B). Therefore, the pose 1 was selected as the preferable pose for CXCL12 binding to CXCR4. More detailed description of docking method is added in the revised manuscript.

- Also, why didn't the authors use any of the experimental structures of chemokines bound to their receptors in their docking? This seems like a valuable piece of information to guide the docking of CXCL12 into CXCR4, but looks like the authors didn't consider it. And why did the re-dock the small-molecule antagonist IT1t to CXCR4, if there is already a crystal structure available?

[Response]

To date, although the respective structures of chemokine CXCL12 and receptor CXCR4 were solved, the complex structure for CXCL12 bound to CXCR4 is still not available. The full-length CXCR4 was constructed by combining crystal CXCR4 structure and CXCL12-bound N-terminal domain of CXCR4 according to the method proposed by Lei Xu *et al.* (ref. 7 in the manuscript). For CXCL12 docking to CXCR4, to increase the accuracy of docking, the TM region of the receptor was blocked and the ligand binding residues from S6-A21 were specified according to previous experiments (ref. 6,7,21 in the manuscript). In addition, we compared our CXCL12–CXCR4 model with other previous CXCL12–CXCR4 models and showed that our model was similar to Volkman's proposed model (ref. 17 in the manuscript) (RMSD= 4.5 Å). We also superposed our model with the complex of viral chemokine vMIP-II bound CXCR4 and the RMSD is 4.3 Å, indicating that our docking results are reasonable.

Although the solved crystal structure of CXCR4 is bound with small-compound IT1t, to validate our docking program for small compound, IT1t was redocked to modeled full-length CXCR4. The preferable docking pose of IT1t to CXCR4 was similar to the solved crystal structure of CXCR4 with IT1t bound and the RMSD of superposition of the two structures was 0.59 Å (Figure S2B).

- The authors claim that their microsecond-scale simulations might have allowed

CXCR4 to reach an intermediate state. While it is difficult to judge what the authors exactly observed without analyzing the trajectories, the kinetics of CXCR4 activation are several orders of magnitude slower. Specifically, FRET measurements have revealed that CXCL12 binding to CXCR4 results in structural rearrangements within the transmembrane domains of the receptor in aprox. 600 ms, of rearrangements between CXCR4 and the G protein in aprox. 1 second and G protein activation in aprox. 4 seconds.

- Perpina-Viciano, C., Isbilir, A., Zarca, A., Caspar, B., Kilpatrick, L. E., Hill, S. J., ... Hoffmann, C. (2020). Kinetic analysis of the early signaling steps of the human chemokine receptor CXCR4. Molecular Pharmacology. DOI: 10.1124/mol.119.118448.

The authors should be more cautious about suggesting that their microsecond-scale simulations allow the receptor to reach intermediate active states.

[Response]

The authors thank for reviewer's comments. For CXCL12-bound CXCR4 system, the MD simulations started from CXCL12 binding to inactive CXCR4, and the conformational changes were evaluated during the dynamic simulations. Although some molecular switches, and conformational changes were found during our microsecond-scale MD simulations, the state which our simulation reached might not be a real intermediate state as compared to the recent work by Perpina-Viciano et al. Even so, previous studies also proposed that there are several intermediate states during the GPCR activation from inactive state to active state (ref. 30 in the manuscript). Our simulations might reach a state toward activation and we corrected the description in the revised manuscript.

While the conclusions of this work seem reasonable –albeit already established in several other GPCRs– I don't think the data presented in the manuscript fully support them. The authors performed medium-length simulations in simplified systems (without providing enough justification on their choices) and they probably observed some rearrangements, as expected in a molecular dynamics simulation. Then, it seems that they simply assigned these trends to already observed phenomena. I can't see a well-thought justification on how the authors reach their conclusions from their simulations.

[Response]

The authors thank for reviewer's suggestions. We revise the statements of conclusions in the revised manuscript.

Reviewers' comments:

Reviewer #1 (Remarks to the Author):

The authors almost finished revisions according to our previous comments. However, there are still some unclear descriptions and lack of some statements, leading to the difficulty in reading and understanding. Further revisions should be required before acceptance.

1. In line 109-110, the authors wrote "CXCL12 docked to mCXCR4 (L2446.40P and L2466.42P) which the two residues were critical in CXCR4 signaling mentioned in previous mutagenesis experiments". But, what specific roles? favor activation or inactivation? Or others? More experimental statements should be given in order to better correlate the computational result with the experimental findings.
2. It is still unclear how the author measured the displacement distance of the residue 6.30, the distance between two residues in TM3 and TM6? Or others? The authors should add the related descriptions and explanations. Furthermore, it is not easy to observed from Figure 3B that TM6 is moving outwards. Recommending to provide an intracellular perspective view and add the mark of the distance that TM6 moves outward in the Figure.
3. Although the author added the analysis of TM6 kind angle to further observe the TM6 changes with time, it is unclear what are the criterions of the TM6 kink angle chosen for active and inactive states. It should be added in the text or figure. In addition, there is lack of descriptions regarding the calculation of the TM6 kink angle and explanations.
4. The section of "materials and methods" is lack of many computational details. It is unclear how the authors calculated these parameters, for example, the tools for visualizations, the calculations of residue distance, hydrogen bonding network and the water density maps.

Reviewer #2 (Remarks to the Author):

All of the points that I raised in the first review process have been well clarified. I think the manuscript is significantly improved, and is now ready for publication.

Reviewer #3 (Remarks to the Author):

The authors have provided more details about the modeling and simulation methods and have clarified many of the points in the manuscript that I found unclear. The text has also improved and is more readable.

Reply to Reviewer 1

The authors thank reviewer for the very helpful comments. Replies to the comments are given in the following. Also, the changes made in the manuscript are highlighted in yellow in the revised manuscript.

Reviewer #1 (Remarks to the Author):

1. In line 109-110, the authors wrote “CXCL12 docked to mCXCR4 (L2446.40P and L2466.42P) which the two residues were critical in CXCR4 signaling mentioned in previous mutagenesis experiments”. But, what specific roles? favor activation or inactivation? Or others? More experimental statements should be given in order to better correlate the computational result with the experimental findings.

[Response]

The authors thank for reviewer’s suggestion. Previous mutagenesis experiments indicated that two residues were deemed critical in CXCR4 signaling based on the substitution to proline (L244^{6.40}P and L246^{6.42}P), which suggested proline mutation in the region can eliminate the downstream signaling without altering extracellular structure or ligand binding (ref. 4,9,21 in the manuscript). The experiments indicated that the two mutations significantly reduced calcium flux and did not affect the CXCL12 binding, which caused the mutant CXCR4 toward inactivation. In our study, to compare with these mutagenesis experiments, we built the mutant CXCR4 with the two residues mutated as proline (L244^{6.40}P and L246^{6.42}P). For CXCL12-bound mutant CXCR4 system, the intercellular half region of TM6 tilted inward to shrink the cytoplasmic binding region for G_i protein and blocked the internal water flow when compared to CXCL12-bound CXCR4 system. The simulation results provided the atomic-level insight and were consistent with these previous experiments and suggested that the conformation of intercellular half region of TM6 is important for signal transmission. We add more experimental statements in the revised manuscript.

2. It is still unclear how the author measured the displacement distance of the residue 6.30, the distance between two residues in TM3 and TM6? Or others? The authors should add the related descriptions and explanations. Furthermore, it is not easy to observed from Figure 3B that TM6 is moving outwards. Recommending to provide an intracellular perspective view and add the mark of the distance that TM6 moves outward in the Figure.

[Response]

The authors thank for reviewer’s suggestion. As mentioned in the manuscript, when GPCRs are activated, the intracellular region of TM6 moves outward to enlarge the

cytoplasmic region of GPCR for G-protein coupling. To clearly represent the outward movement of TM6, we superposed the CXCR4 structures at different time frames (0, 500, 1000, 1500, 1800 ns) and measured the C α atom movement of the intracellular end residue of TM6 (K234^{6.30}) shown in Figure 3B. The representation was also used in other GPCR systems. We replace the side-view superposition of CXCR4s in Figure 3B with the intracellular view superposition of CXCR4s based on the reviewer's suggestion. We also add more related descriptions in the revised manuscript.

3. Although the author added the analysis of TM6 kink angle to further observe the TM6 changes with time, it is unclear what are the criterions of the TM6 kink angle chosen for active and inactive states. It should be added in the text or figure. In addition, there is lack of descriptions regarding the calculation of the TM6 kink angle and explanations.

[Response]

The authors thank for reviewer's suggestion. The kink angles of TM6 (the angle between up half and down half of TM6) with time for various simulation systems were shown in Figure S3F. Three C α atoms of residues (I245^{6.41}, P254^{6.50}, and G258^{6.54}) were selected to measure the kink angle of TM6. The negative kink angle for CXCL12-bound mCXCR4 system meant inward tilt of down half of TM6. Larger down half tilt movement of TM6 means the smaller kink angle of TM6. The larger kink angles of TM6 were found in apo CXCR4 and IT1t-CXCR4 systems, similar to the inactive CXCR4 crystal structure. We add the criterions of TM6 kink angles of inactive CXCR4 crystal structure (PDB: 3ODU) and active β_2 AR-Gs complex structure (PDB: 3SN6) in the revised Figure S3F and detailed description regarding the calculation of TM6 kink angle in the revised manuscript.

4. The section of “materials and methods” is lack of many computational details. It is unclear how the authors calculated these parameters, for example, the tools for visualizations, the calculations of residue distance, hydrogen bonding network and the water density maps.

[Response]

According to the reviewer’s suggestion, we add more computational details and analyses in the revised manuscript, including the calculations of residue distance, hydrogen bonding network, and water density maps.

REVIEWERS' COMMENTS:

Reviewer #1 (Remarks to the Author):

The authors already well clarified all of the points that I raised in the second review process. The manuscript is further improved and more readable. So,I recommend its publication.